# Spatial transcriptomics reveals unique gene expression changes in different brain regions after sleep deprivation

Yann Vanrobaeys [1,2,3], Zeru J. Peterson [2,4], Emily. N. Walsh [2,3,5], Snehajyoti Chatterjee [2,3], Li-Chun Lin[2,3,6], Lisa C. Lyons[7], Thomas Nickl-Jockschat [2,3,4] ✉ & Ted Abel [2,3] ✉

Sleep deprivation has far-reaching consequences on the brain and behavior, impacting memory, attention, and metabolism. Previous research has focused on gene expression changes in individual brain regions, such as the hippocampus or cortex. Therefore, it is unclear how uniformly or heterogeneously sleep loss affects the brain. Here, we use spatial transcriptomics to define the impact of a brief period of sleep deprivation across the brain in male mice. We find that sleep deprivation induced pronounced differences in gene expression across the brain, with the greatest changes in the hippocampus, neocortex, hypothalamus, and thalamus. Both the differentially expressed genes and the direction of regulation differed markedly across regions. Importantly, we developed bioinformatic tools to register tissue sections and gene expression data into a common anatomical space, allowing a brain-wide comparison of gene expression patterns between samples. Our results suggest that distinct molecular mechanisms acting in discrete brain regions underlie the biological effects of sleep deprivation.

Sleep deprivation is a growing problem that affects more than one-third of adults in the U.S. and more than 70% of teenagers and adolescents[1]. Loss of sleep impacts cognition, attention, and metabolism[2–5]. These processes are mediated by distinct neural circuits in specific brain regions–the hippocampus, the cortex, and hypothalamus, respectively. Sleep and circadian rhythm disorders have also been linked to the increased incidence and accelerated progression of neurodegenerative diseases, including Alzheimer's disease[6–10]. Given the serious consequences of sleep loss for individuals and the interaction of sleep deprivation with many diseases, it is important to understand the cellular and molecular consequences of sleep deprivation. To this end, we have used non-biased spatial

transcriptomics to define whether sleep loss has distinct molecular impacts on specific brain regions.

Sleep deprivation impacts protein synthesis and gene regulation through many mechanisms including alterations to epigenetic regulation, transcription, and mRNA processing[11–19]. Estimates suggest that up to 10% of cortical transcripts are regulated with sleep/wake cycles, particularly by the length of time awake[20–22]. In the hippocampus, prolonged wakefulness causes changes in the expression of genes associated with RNA splicing, cell adhesion, dendritic localization, the synapse, and the postsynaptic membrane[11,13,23,24]. However, the brain is highly heterogeneous and subserves many different functions; as brain regions and circuits differ in their roles, they may

[1]Interdisciplinary Graduate Program in Genetics, University of Iowa, 357 Medical Research Center Iowa City, Iowa, IA, USA. [2]Iowa Neuroscience Institute, Carver College of Medicine, University of Iowa, 169 Newton Road, 2312 Pappajohn Biomedical Discovery Building, Iowa City, IA, USA. [3]Department of Neuroscience and Pharmacology, Carver College of Medicine, University of Iowa, 51 Newton Road, 2-417B Bowen Science Building, Iowa City, IA, USA. [4]Department of Psychiatry, University of Iowa, Iowa City, IA, USA. [5]Interdisciplinary Graduate Program in Neuroscience, University of Iowa, 356 Medical Research Center, Iowa City, IA, USA. [6]Department of Neurology, University of Iowa, Iowa City, IA, USA. [7]Program in Neuroscience, Department of Biological Science, Florida State University, Tallahassee, FL, USA. ✉e-mail: thomas-nickl-jockschat@uiowa.edu; ted-abel@uiowa.edu

differ dramatically in their response to sleep loss, and observations from one brain region may not be generalized to the whole brain.

Recent technological advances in genome-wide spatial transcriptomics offer enormous potential for providing detailed molecular maps that overcome limitations associated with single cell or single nuclear RNA sequencing (sc/snRNA-seq) and microscopy-based spatial transcriptomics methods[25]. This approach has been successfully used to generate detailed datasets and cell-type specific gene expression signatures[26–29], but it has not yet been used to profile changes in gene expression across multiple brain regions after experience. A further challenge for data analysis is that a significant hurdle remains in terms of finding a strategy to align the brain regions across slices from multiple subjects or from independent experiments for data integration in multi-sample analyses. To investigate gene expression changes within the adult mouse brain after sleep deprivation, we used the 10x Genomics Visium platform, a barcoding-based, transcriptome-wide approach that generates spatial maps of gene expression. We collected gene expression data from each major brain region across a coronal brain slice, enabling us to profile multiple brain regions simultaneously. Using this technique, we were also able to get detailed, subregion and layer specific gene expression changes within the hippocampus and cortex. Finally, we present an alternative to a region-of-interest type of analysis by registering multiple slices into a common space using the Common Coordinate Framework (CCF) from the Allen Brain Atlas[30], thus adjusting for differences in the alignment of brain tissue sections and allowing for a comparison between samples. These data and analytical approaches provide a scientific resource for the neuroscientific community, and they demonstrate the diverse impact of sleep loss on gene expression across the brain.

## Results

Using spatial transcriptomics, we profiled spatial gene expression in coronal brain slices from sleep-deprived (SD) or control (non-sleep deprived (NSD)) adult male mice. Each coronal section covered between 1736 and 3103 spots on the Visium slides (Supplementary Data 1). We sequenced each sample to a median depth of 2.26E + 08 (interquartile range 2.10E + 08–2.37E + 08), which corresponded to a mean of 93245 reads and a mean of 5978 genes per spot (Supplementary Data 1). Importantly, we were able to detect over 21,000 genes for each sample (Supplementary Data 1). However, the individual number of genes detected in each spot is three to four times lower than the total number of expressed genes detected due to cell-type and brain-region specific differences in gene expression (Supplementary Fig. 1). We note that these rates are analogous to snRNA-seq and scRNA-seq data using the 10x Genomics Chromium platform, where a "cell" barcode on the Chromium platform corresponds to a "spatial" barcode on the Visium platform. However, unlike snRNA-seq data, which contains high numbers of intronic reads that map to immature transcripts, we found strong enrichment of mature messenger RNAs with high mean rates of exonic alignments (mean: 88.3%; IQR: 87.7–89.4%) (Supplementary Data 1).

We first generated region-enriched expression profiles for the samples from each condition (Fig. 1A–C and Supplementary Fig. 2). As expected, this approach predicted brain regions with high reliability (Fig. 1B) and recapitulated the brain regions from the reference coronal mouse Allen brain atlas[31] (Fig. 1C). Each brain region was characterized by specific transcriptional signatures and unsupervised clustering of these region expression profiles revealed distinct clusters (Fig. 1D) and top biomarkers (Fig. 1E). Supplementary Table 2 reports the expression levels for all genes with no filters. Due to the transcriptional similarity of the basomedial and basolateral amygdalar subnuclei with the subnuclei of the amygdalar medial area, and the spatial proximity and similarity with the allocortex, UMAP clustering reports the striatum-like amygdalar nuclei and the allocortex as repeated clusters with slightly different transcriptional signatures

depending upon the grouping of the basomedial and basolateral subnuclei (Fig. 1D). We merged anatomically adjacent spots from the two clusters to generate the labeled brain regions. Together, these results highlight the ability of spatial transcriptomics to achieve high-resolution expression profiling across the mouse brain.

## Sleep deprivation exerts differential effects on transcriptional activity in each brain region

Sleep deprivation affects different brain functions ranging from cognition and affective processing that each rely upon distinct neuronal circuits[17,22–24,32–35]. However, little is known about how sleep deprivation alters transcriptomic activity in individual brain regions, as bulk sequencing approaches inevitably average out regionalized effects. To address this problem, we performed differential gene expression analysis in each of the brain regions identified in the coronal sections (Fig. 1). In the analysis, significant differentially expressed genes (DEGs) were identified using two filters, an FDR of 0.001 and an absolute fold change threshold of 1.2. Significant DEGs for each brain region are reported in Supplementary Table 3. After filtering, we found that the hippocampal region had the greatest number of significant DEGs affected by sleep loss (592 DEGs), followed by the neocortex (401 DEGs), the hypothalamus (266 DEGs), and the thalamus (113 DEGs) (Fig. 2A). Some of these DEGs, such as *Rbm3*, *Hspa5*, and *Srsf5*, have been previously shown to be affected after sleep deprivation in our previous studies of the hippocampus[11,36,37] and in studies of other brain regions[13–15,17,21,34,38,39].

The molecular functions of the DEGs showed region-specific differences (Fig. 2B–E and Supplementary Data 4). For the hippocampal region, many molecular functions related to RNA processing were enriched (Fig. 2B). For the neocortex, molecular functions related to protein kinase activity, GTPase activity, ubiquitin ligase activity, and DNA-binding transcription factor binding were enriched (Fig. 2C). The DEGs in the hypothalamus were enriched for molecular functions related to neuropeptide and hormone activity, as well as glutathione transferase and peroxidase activity (Fig. 2D). Finally, the DEGs in the thalamus were enriched for the Myogenic Regulatory Factor (MRF) binding molecular function (Fig. 2E). Surprisingly, ~98% of the DEGs in the hippocampal region were significantly downregulated whereas ~96% of the DEGs in the neocortex were significantly upregulated (Fig. 2B, C and Supplementary Fig. 3).

We next investigated how many of those total DEGs are uniquely affected in each brain region by analyzing the degree of overlap between the DEGs in the brain regions that had at least 50 DEGs affected by sleep deprivation (Fig. 2F). Although there were many connections between different brain regions, the majority (50–83%) of the DEGs were specifically affected in their respective brain region. Of the 592 DEGs found in the hippocampal region, 489 were exclusively affected in the hippocampal region (489/592 DEGs), 306/401 in the neocortex, 199/266 in the hypothalamus, 56/113 in the thalamus, and 33/66 in the striatum-like amygdalar nuclei (Supplementary Data 3). Interestingly, only 35 DEGs were found to be in common between the neocortex and the allocortex, resulting in one significantly enriched pathway: protein kinase inhibitor activity (Supplementary Fig. 4). All other sets of common DEGs (Supplementary Data 5) were too few in number to reveal enriched molecular functions.

## Hippocampal subregions are differentially impacted by sleep deprivation

As our results here and previous studies have demonstrated, the hippocampus is highly susceptible to the effects of acute sleep deprivation[11,13,24,36,37]. This brain region is comprised of several substructures—CA1, CA2, CA3, and the dentate gyrus (DG)—each with different functions in learning and memory[40–45]. We performed a deconvolution of the CA1 pyramidal layer and the dentate gyrus (DG) granule cell layer using a reference scRNA-seq whole hippocampus

mouse dataset from the Allen Brain Atlas[46] (Fig. 3A) and were able to distinguish the areas CA2 and CA3 pyramidal layers based on spatial topography. Similarly, because the dendritic layers of CA1 are known to undergo structural changes following sleep deprivation[47–49], we also used spatial topography to define and include the stratum radiatum and oriens layers of CA1 in our analysis (Fig. 3B). Differential gene expression analysis in each hippocampal subregion revealed unique gene expression changes and molecular functions enriched that were specific to a subregion (Fig. 3C, Supplementary Data 6 and 7). Of the DEGs identified in each region, 51/62 DEGs were uniquely affected in CA1, 34/41 in DG, 53/61 in stratum radiatum, and 4/4 in stratum oriens. The CA1 pyramidal layer and stratum radiatum were most impacted by sleep deprivation, with the most DEGs and unique DEGs of the areas examined. Stratum radiatum had 53 unique DEGs enough to enrich the cyclin-dependent protein serine/threonine kinase activity, as well as the pyramidal CA1 cells with their 51 unique DEGs that enriched the glutamate receptor binding. Interestingly, there were no genes significantly affected in the combined CA2 and CA3 pyramidal layers after sleep deprivation. This finding supports other observations that CA1 and the DG are impacted by sleep deprivation while area CA3 is less affected[37,49].

## Sleep deprivation causes layer-specific transcriptional changes in the cortex

The neocortex was the second-most impacted by sleep deprivation (Fig. 2A). The cortex comprises of different layers that each are involved in various functions of receiving, integrating, and outputting information[50]. To understand how sleep deprivation differently impacts the layers of the cortex, we examined the gene expression profiles within each cortical layer. We performed a deconvolution of the spatial datasets by integrating them with a reference scRNA-seq dataset of ~14,000 adult mouse cortical cell taxonomy from the Allen Institute[51]. This allowed us to identify the layers of the neocortex based on the prediction score in each spot (Fig. 4A) and perform differential gene expression analyses in each layer (Supplementary Data 8 and Supplementary Fig. 5). Layers 2/3 and 5 are the most transcriptionally affected after sleep deprivation with 222 and 225 significant DEGs, respectively (Supplementary Data 9). Differential gene expression analysis in each cortical layer revealed distinct gene expression changes and molecular functions that were uniquely enriched in certain layers (Fig. 4B), which may relate to the differential function of these layers in intracortical processing and cortical output. Layer 5, which contains neurons that are the main output of the cortex, had 174 unique DEGs that included molecular functions related to sterol binding, cyclic adenosine monophosphate (cAMP) binding, structural constituents of the postsynapse, and ion channel regulator activity. Layer 2/3, which functions largely in information processing within the cortex, had 149 unique DEGs that included molecular functions related to phosphatase inhibitor activity, adenylate cyclase inhibiting G protein-coupled glutamate receptor activity, and ionotropic glutamate receptor binding.

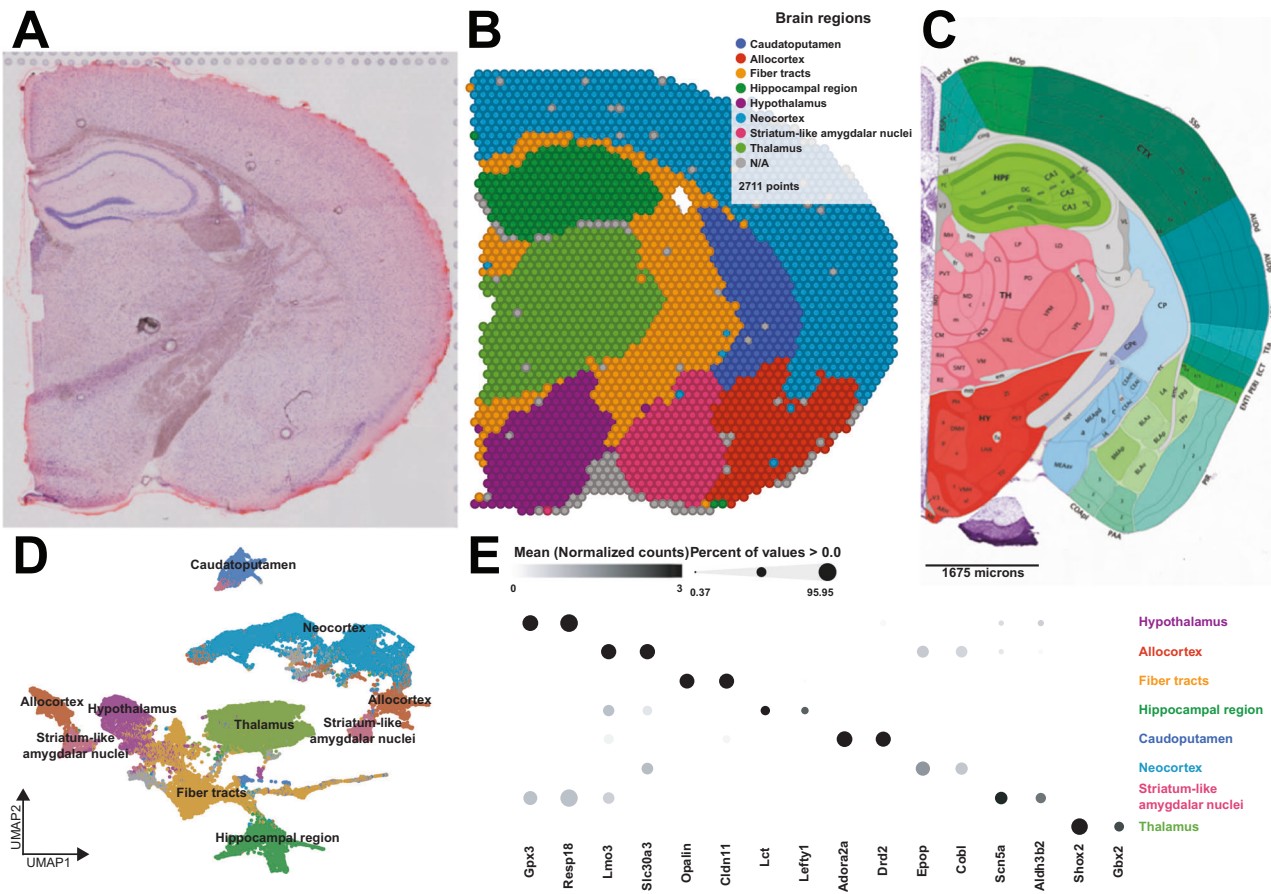

**Fig. 1 | Spatial patterns of gene expression define anatomically distinct brain regions. A** Example of coronal tissue section H&E histology staining from sample 4. This process was repeated for each of the 16 mice. **B** Graph-based cluster identification from spot-level (2711 spots) of sample 4. Each spot is colored based on the transcriptional signature computed from 20 principal components using Louvain clustering algorithm. The brain regions are labeled in the colored legend. This process was repeated for each of the 16 mice. **C** Adapted screenshot of the Allen Reference Atlas−Mouse Brain (coronal section image 72 of 132, position 285, http://atlas.brain-map.org/). **D** UMAP plot based on the transcriptional signature of each spot. **E** Bubble plot of the most significant computed biomarkers for each brain region. The bubble chart shows the expression level of biomarkers in each brain region. Bubble diameters are proportional to the percentage of spots that show expression of the biomarker. For each brain region, two significant biomarkers are displayed.

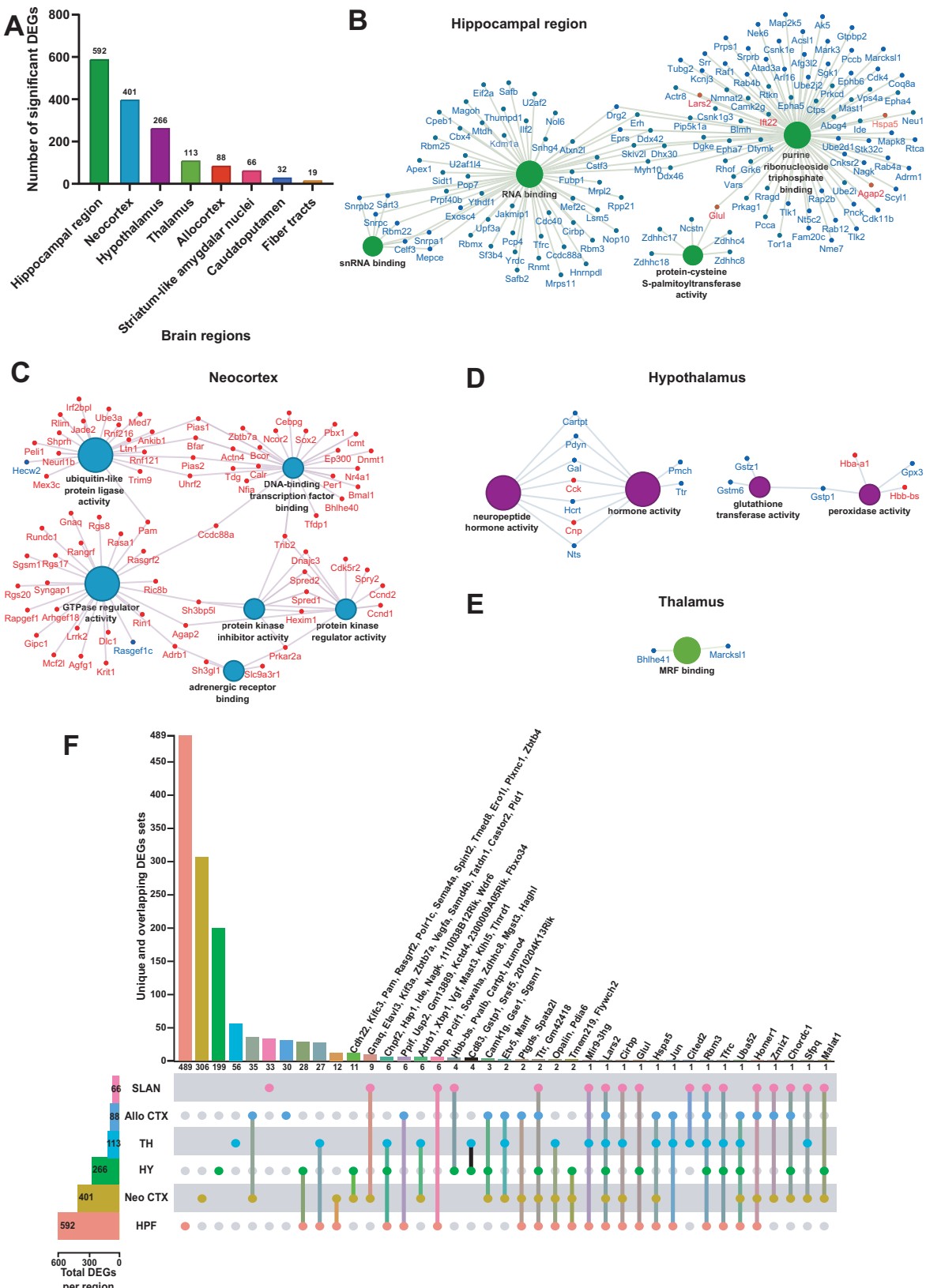

## Registration of slices to a common anatomical reference space via the Spatial Transcriptomics Analysis Tool (STAnly) allows the unrestricted analysis of transcriptomic data across entire brain slices

Our deconvolution approach (used in Figs. 1–4) subdivides a given brain slice into different larger brain regions based on their transcriptomic activity. Although this is a powerful tool to analyze spatial gene expression changes, it inevitably comes at the price of a loss of spatial resolution, as this approach necessarily pools over larger brain regions, and requires a prior biological knowledge of cell type-specific gene expression profiles. To address this loss of spatial resolution, we established an analysis tool (Spatial Transcriptomics

**Fig. 2 | The hippocampal region is the brain region the most transcriptionally affected after sleep deprivation. A** Histogram representing the number of significant differentially expressed genes (DEGs) across each brain region previously identified. Molecular functions enriched from the significant DEGs in the hippocampal region (**B**), neocortex (**C**), hypothalamus (**D**), thalamus (**E**). A gene is significant if its FDR step-up <0.001 and its log2fold-change ≥|0.2|. The size of the circle for each enriched molecular function is proportional to the significance. Only molecular functions with a corrected $p < 0.05$ are displayed (two-sided hypergeometric test, Bonferroni step down). The DEGs within these molecular functions are color coded to show whether they are downregulated (blue) or upregulated (red).

**F** UpSet plot of interactions between each brain region that have more than 50 significant DEGs (fiber tracts and caudatoputamen excluded). The number of DEGs submitted for each brain region is represented by the histogram on the left (0–600 range). Dots alone indicate no overlap with any other lists. Dots with connecting lines indicate one or more overlap of DEGs between brain regions. The number of DEGs in a specific list that overlap is represented by the histogram on the top. For spatial expression patterns with smaller numbers of DEGs, we were able to list the gene names above their respective histogram. Genes are labeled for the smallest lists. HPF Hippocampal Formation, Neo CTX Neocortex, HY Hypothalamus, TH Thalamus, Allo CTX Allocortex, SLAN Striatum-like amygdalar nuclei.

ANaLYsis (STANLY) that aligns dots from multiple samples from different animals into one common anatomical reference space, the Common Coordinate Framework (CCF) of the Allen Mouse Brain Atlas[31], thus allowing a dot-by-dot comparison of the transcriptome in an unrestricted inference space (Fig. 5A). To account for different numbers of spots across slices, we generated "digital spots" in this same coordinate system to allow a statistical comparison across. Using this method, we detected at least 18,893 genes in all sample slices for changes in expression between NSD and SD. Of these, 413 genes (Supplementary Data 10) were significantly differentially expressed, with 150 genes showing an upregulation in all significant spots, 22 showing downregulation in all significant spots, and 256 showing a combination of up and down regulation across the sample space. These DEGs include previously described upregulated genes like *Per1* (Fig. 5B), *Nr4a1* (Fig. 5C), *Homer1* (Fig. 5D), and *Arc* (Fig. 5E), which showed localized increases in the neocortex, as well as downregulated genes like *Rbm3* (Fig. 5F) and *Cirbp* (Fig. 5G), which showed hippocampus-specific changes, similar to those seen in our deconvolution approach. Using ToppGene[52], we found the top five enriched mouse phenotypes were related to abnormal synaptic transmission (83 DEGs), abnormal synaptic physiology (83 DEGs), abnormal learning/memory/conditioning (84 DEGs), abnormal cognition (84 DEGs), and abnormal CNS synaptic transmission (75 DEGs) across the whole coronal slice. GO-molecular function (GO:MF) enrichment analysis showed similar functions enriched in previously identified brain region such as RNA binding (found in the hippocampus), Ubiquitin-like protein ligase activity, GTP binding, kinase activity (found in the neocortex), and neuropeptide and hormone activity (found in the hypothalamus) (Supplementary Fig. 6 and Fig. 3).

We wanted to test the power of STANLY and spatial transcriptomics for subregional analysis within the hippocampus, particularly area CA1 and the dentate gyrus. Both area CA1 and the dentate gyrus were treated as binary masks, with dots situated within these regions included into our analyses. We then conducted a two-sample t-test at each spot within these masks as shown in Supplementary Fig. 7. Because we were only investigating one gene, *Arc*, we performed our p-value correction using the number of spots being tested, which was 140, giving us a threshold of $p < 0.00075$. Using this p-value, we found that sleep deprivation significantly increased the mRNA expression of the activity-dependent immediate early gene *Arc* in area CA1 ($p < 0.0005$), whereas there was little change or even slightly decreased expression of *Arc* in the dentate gyrus. To validate the reliability of our STANLY analyses, using samples from independent sleep deprivation experiments, we performed in situ hybridization using RNAscope (ACD) for *Arc* expression in the hippocampus. The RNAscope analysis revealed that acute sleep deprivation significantly increased Arc mRNA positive cells in the CA1, while there was no change in the DG (Supplementary Fig. 7). *Arc* shows increased expression in the hippocampus following acute sleep deprivation[11,13]. Moreover, *Arc* has also been identified as having differential expression in the subregions of the hippocampus after sleep deprivation with significantly increased levels of Arc in the CA1 and no change or decreased expression in the dentate gyrus[53]. Thus, the results from STANLY analysis comparing subregions within the hippocampus are

consistent with our in situ hybridization experiments and with previously published results. These results validate the power of STANLY and spatial transcriptomic approaches to identify spatially restricted changes in gene expression.

## Discussion

The identification of cell-type specific transcriptomic signatures has been invaluable in distinguishing subclasses of cell types in the brain[54] and has provided insights into brain disorders such as epilepsy, autism, and Alzheimer's disease[55–57]. However, the lack of spatial information associated with single cell transcriptomics represents a significant obstacle[58,59], especially in an organ as complex as the brain. Spatial transcriptomics combines a spatial barcode of RNA transcripts with near single cell sequencing resolution providing a major advance for understanding gene regulation across brain regions. However, the recent development of this technology means that it is largely untested for the analysis of differential gene expression. Here, we used this technique to examine the important problem of how acute sleep deprivation affects gene expression across brain regions. The effects of sleep deprivation on public health, and as a risk factor increasing the susceptibility and incidence of numerous diseases, necessitate that we utilize and develop techniques that will provide more detailed understanding of the consequences of sleep loss.

The spatial transcriptomic platform provided sequencing depth comparable to single cell and single nuclear transcriptomic studies in terms of gene number per spot, with the advantage of enriching mature RNA transcripts (Supplementary Data 1). Potentially, the clustering of a small number of cells in the spots of the slides allows for a greater sequencing of mature cytoplasmic RNA molecules, compared to the nuclear mRNA that contains immature RNAs still being processed. This technique allowed us to anatomically distinguish individual brain regions by aligning brain regions with the reference mouse Allen brain atlas, where we found that individual brain regions showed distinct transcriptional profiles after acute sleep deprivation. Individual cell types clustered within a brain region similar to single cell transcriptomic studies (Fig. 1). Thus, these results demonstrate the comparability of spatial transcriptomics to the resolution of single-cell approaches with the added power of simultaneous brain-wide investigation and additional spatial information.

Given the recent development of the spatial transcriptomics platform, we employed both a relatively large number of samples for a transcriptomics study and a highly conservative statistical analysis using an FDR of 0.001 to determine differential gene expression in individual brain regions following acute sleep deprivation. Using a conservative FDR-corrected $p < 0.001$ as the threshold, we identified fewer differentially expressed genes in the hippocampus and the cortex compared to other transcriptomic studies[11,13,21]. As our sleep deprivation method and protocols were similar to other studies[11,13], we believe that differences arise from our use of a highly conservative statistical approach to avoid false positives due to the large sample size (i.e., the number of spots in each slice). However, we recognize that our conservative approach may also result in false negatives, so all gene expression data is included in the Supplementary Tables and available through GEO (GSE222410). Importantly, all samples were collected at

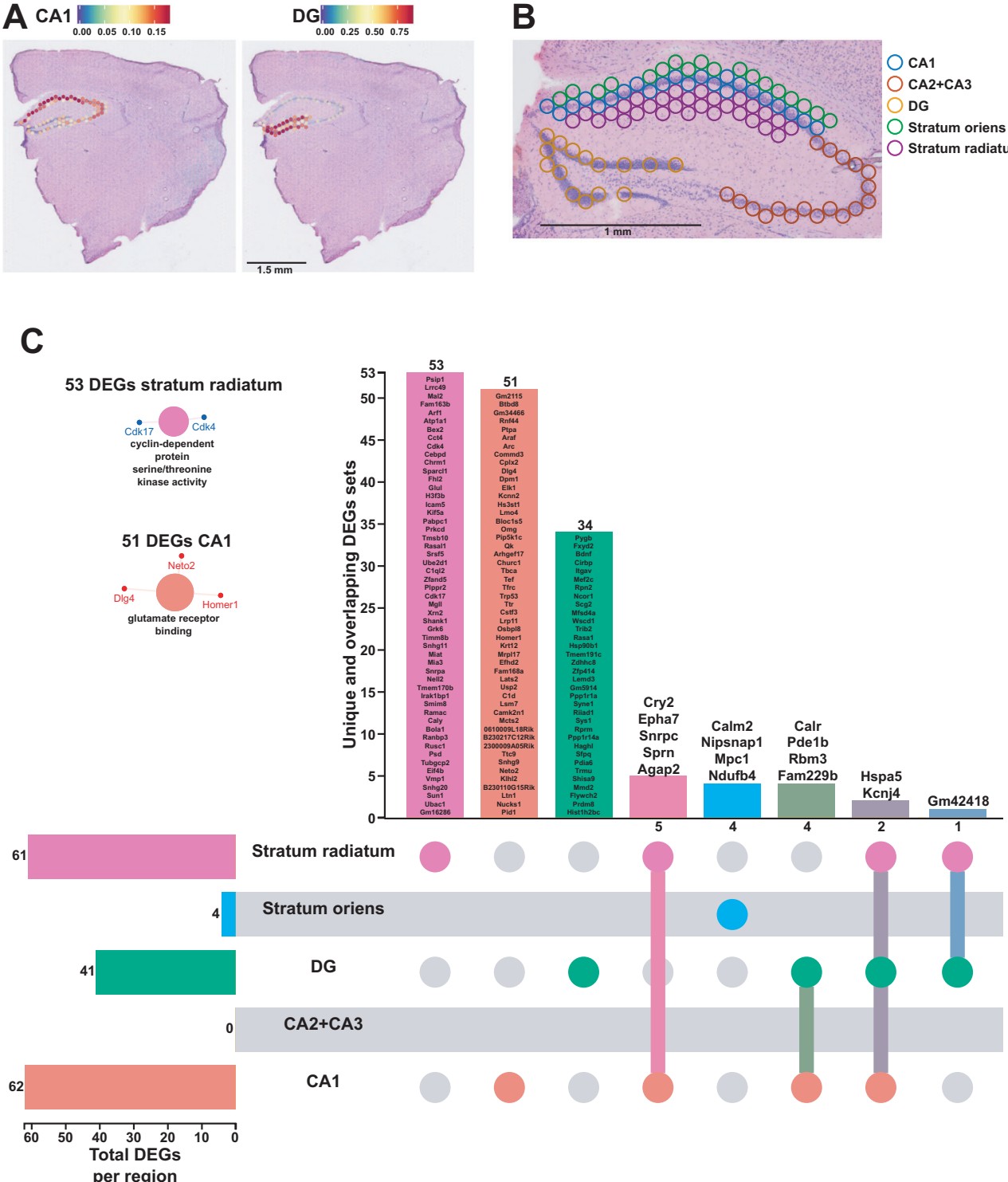

**Fig. 3 | Each hippocampal subregions displays a unique transcriptional impact of sleep deprivation. A** Prediction score of the deconvolution step for each of the 2085 spots of a representative example slice for CA1 pyramidal layer and dentate gyrus (DG) granule cells are represented with the color legend from blue to red. The rest of the subregions were selected based on biological knowledge using anatomical structures apparent on the H&E staining images. **B** Example of identified hippocampal subregions on sample 16. **C** UpSet plot of interactions between each hippocampal subregion. The number of differentially expressed genes (DEGs) submitted for each subregion is represented by the histogram on the left (0–62 range). A gene is significant if its FDR step-up <0.1 and its log2fold-change ≥ |0.2|.

Dots alone indicate no overlap with any other lists. Dots with connecting lines indicate one or more overlap of DEGs between hippocampal subregion. The number of DEGs in a specific list of overlap is represented by the histogram on the top. Genes are labeled for the smallest lists. The unique lists of 53 DEGs and 51 DEGs for stratum radiatum and CA1 pyramidal cells respectively enriched specific molecular functions displayed on the left. The size of the circle for each enriched molecular function is proportional to the significance. Only molecular functions with a corrected $p < 0.05$ are displayed (two-sided hypergeometric test, Bonferroni step down). A gene is considered significant if FDR < 0.001 and log2fold change > |0.2|.

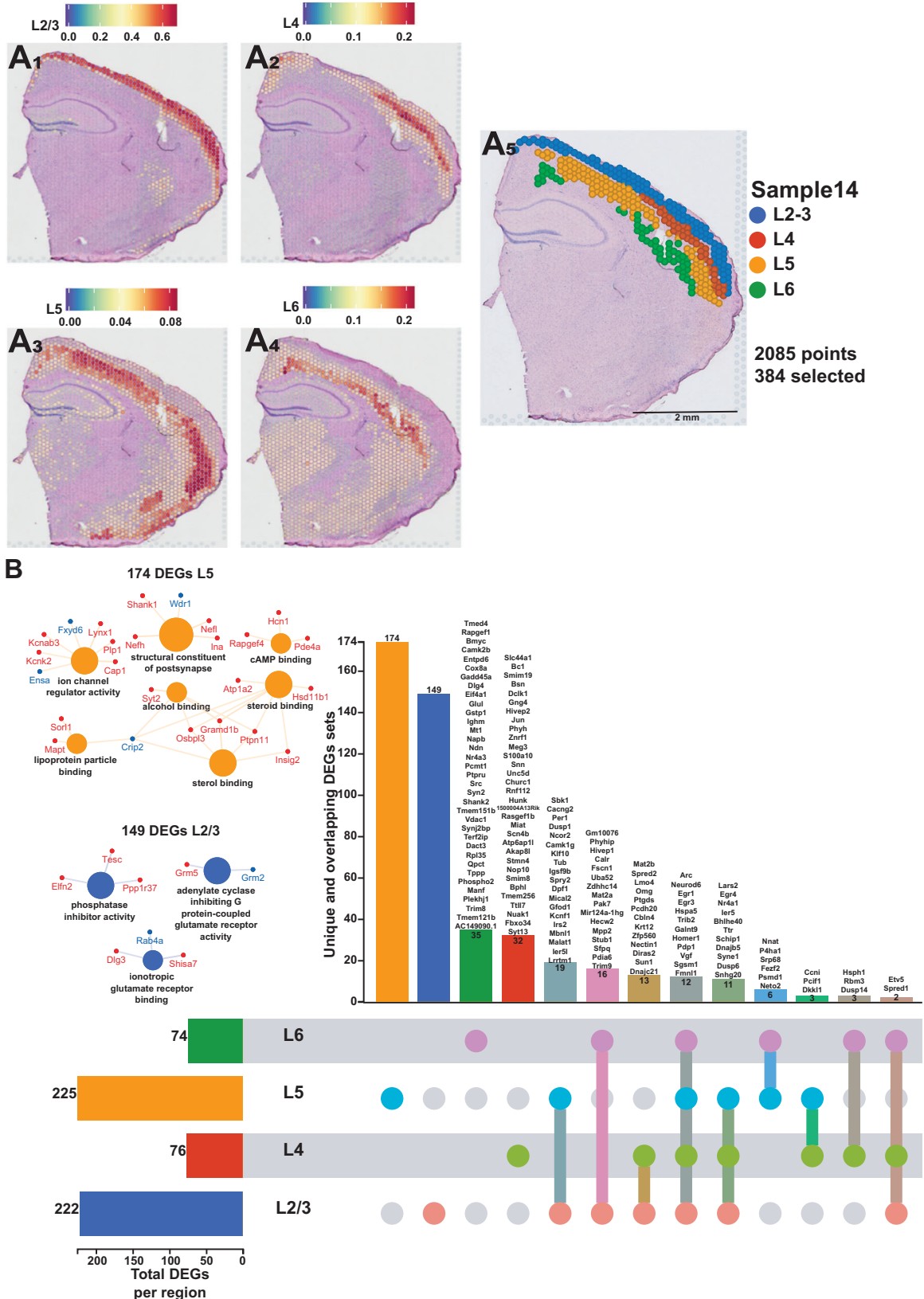

the same time of day as the circadian clock has independent effects on transcription[60,61]. In the current study, we were unable to investigate potential asymmetries in the response of the left and right hemisphere to the impacts of acute sleep deprivation, as we were limited by the size of the spatial transcriptomic slides in which only one hemisphere can fit within the bounds of the slide. In future studies, we plan to use

spatial transcriptomics to investigate more nuanced impacts of sleep deprivation, such as interhemispheric differences.

We found that acute sleep deprivation had the greatest impact on gene regulation in the hippocampus, neocortex, hypothalamus and thalamus (Fig. 2A). Interestingly, this conservative approach strongly illustrated heterogeneity of brain regions in response to sleep

**Fig. 4 | Each cortical layer of the neocortex displays a unique transcriptional impact of sleep deprivation. A** Prediction score of the deconvolution step for each of the 2085 spots of a representative example slice for each cortical layer are represented with the color legend from blue to red: layer 2–3 (**A₁**), layer 4 (**A₂**), layer 5 (**A₃**), layer 6 (**A₄**). We can distinguish between distinct sequential laminar excitatory neurons layers on the aggregated profile (**A₅**). **B** UpSet plot of interactions between each deconvoluted cortical layers of the neocortex. The number of differentially expressed genes (DEGs) submitted for each layer is represented by the histogram on the left (0–225 range). A gene is significant if its FDR step-up <0.001 and its log2fold-change ≥ |0.2|. Dots alone indicate no overlap with any other lists.

Dots with connecting lines indicate one or more overlap of DEGs between cortical layers. The number of DEGs in a specific list of overlap is represented by the histogram on the top. Genes are labeled for the smallest lists. L2/3 = Layer 2 and 3; L4 = Layer 4; L5 = Layer 5; L6 = Layer 6. The unique lists of 174 DEGs for layer 5 and 149 DEGs for layer 2/3 that enrich specific molecular functions are listed on the left. The size of the circle for each enriched molecular function is proportional to the significance. Only molecular functions with a corrected $p < 0.05$ are displayed (two-sided hypergeometric test, Bonferroni step down). A gene is considered significant if FDR < 0.001 and log2fold change > |0.2|.

deprivation, as we found little overlap in the differentially expressed genes across brain regions (Fig. 2F). Moreover, our results conclusively demonstrate that directional changes in gene expression following acute sleep deprivation vary widely across brain regions; approximately 98% of the differentially expressed genes downregulated in the hippocampus, while the opposite was true in the neocortex, which had approximately 96% of the differentially expressed genes upregulated (Fig. 2B, C). Thus, analysis of gene expression changes after acute sleep deprivation in older studies, in which the entire forebrain was collected, may have masked the nuanced effects of sleep deprivation on gene regulation. The dramatic differences in gene expression across brain regions in response to sleep deprivation also suggests that a single theory to explain the impact of wakefulness on the brain or the function of sleep is unlikely to be satisfactory.

The work presented here establishes the robustness and fidelity of spatial transcriptomics for the determination and analysis of differential gene expression within brain subregions as well as for comparisons of gene expression across the brain. For example, in the hippocampus, we found that acute sleep deprivation significantly reduced gene expression involved in RNA processing similar to what was found in previous research[11]. In the neocortex, upregulation was observed for genes involved in DNA binding and transcription factor activity, protein kinase regulation, GTPase regulation and ubiquitin like protein ligase activity. This upregulation of genes involved in DNA binding and transcription factor activity, such as the transcription factor *Nr4a1*, may explain the greater percentage of upregulated genes found in the neocortex as increased expression of NR4A1 would lead to increased expression of its target genes. Although a smaller number of genes were identified in the hypothalamus and thalamus, they nonetheless indicate significant changes in molecular function[17]. For instance, we found that the most significant alterations in the hypothalamus were for genes associated with neuropeptide and hormone signaling. The differences in the functions and molecular pathways affected in each region may provide key insights into how each structure is related to some of the broader and longer lasting effects of acute sleep deprivation. Importantly, the differentially expressed gene functions we identified in each brain region are consistent with the behavioral effects that have been observed following sleep deprivation and attributed to changes in neuronal function, such as changes in circadian behavior or impairments in long-term memory.

The high density of individually coded spots on the slide grid enabled sub-regional analysis of gene expression between slices from sleep deprived and non-sleep deprived mice when combined with a deconvolution approach using single cell reference data sets from the Allen Brain Atlas for the hippocampus (Fig. 3A) and the cortex (Fig. 4A). Subregional analysis of the hippocampus was done for the CA1, CA2/3 pyramidal cell layers, dentate gyrus granule cell layer, and the stratum oriens and the stratum radiatum which contain diverse populations of interneurons. Although both the stratum oriens and the stratum radiatum contain interneurons, the functions of these two layers are distinct, and receive different anatomical inputs. Given the disparate functions and circuitry of the hippocampal subregions, we predicted that sleep deprivation would result in distinct transcriptional profiles in these subregions. With the decreased number of sample spots in the

analysis of hippocampal subregions, the FDR threshold was lowered to 0.1 for the identification of significant DEGs to reduce the number of false negatives, similar to the FDR used for RNA-seq studies of hippocampal subregions[62]. We found that sleep deprivation induced the largest number of changes in gene expression in the CA1 and stratum radiatum. Surprisingly, there were only four genes affected by sleep deprivation in the stratum oriens, although interneurons within this region have been shown to be plastic and provide input to CA1 pyramidal cells[63]. These results suggest that sleep deprivation has the broadest impact on gene regulation in the excitatory neurons of the hippocampus. This result is consistent with previous research in which manipulations of protein synthesis within hippocampal excitatory neurons ameliorated the impacts of sleep deprivation on hippocampus dependent long-term spatial memory[64]. However, it should be noted that the power of subregional analysis for differential gene expression within the hippocampus may be limited by the number of spots in each subregion. In comparison to the individual layered analysis of the neocortex, there were fewer differentially expressed genes detected in the subregions of the hippocampus (Fig. 3C vs 4B). It is probable that the small number of sample spots and subsequent lack of statistical power for analysis of the CA2 and CA3 subregions resulted in a failure to detect DEGs with some genes reported as false negatives. Future research in which single-cell RNA-seq is combined with spatial transcriptomics could resolve these issues.

We found that within the neocortex, sleep deprivation differentially affected individual cortical layers (Fig. 4B), and that Layers 2/3 and 5 were the most affected by sleep deprivation. Interestingly, changes in gene expression following sleep deprivation were unique for individual layers: more than 65% of the genes were unique in Layer 5 and 75% of the genes in Layer 2/3 were unique. Although the number of genes affected was smaller for Layer 4 and Layer 6, the number of layer specific gene changes for these layers was still approximately 50%. From this we can observe that there are distinct impacts of sleep deprivation on individual cortical layers. Indeed, Layer 2/3 function as corticocortical projections to layer 5 and form a prominent interlaminar pathway to amplify, integrate, distribute and temporarily store information within subsets of neurons[65]. From the Layer 5, pyramidal tract neurons project to multiple targets including ipsilateral striatum, thalamus, subthalamic nucleus and many brainstem and spinal cord regions[66]. The elevated level of response from these two layers highlight how the cortex is adapting in response to sleep deprivation, and these connections may better illustrate why cortical functions and properties are so altered by sleep loss[67].

Spatial transcriptomics provides a potentially powerful approach for large scale comparisons of gene expression across multiple conditions or disease states. For the full capability of spatial transcriptomics to be realized, it is necessary to develop the analysis tools for the alignment of spatial transcriptomic data sets into a common anatomical reference space to allow an unrestricted comparison of gene expression between samples. To further this goal, we pioneered the adaption of bioinformatic tools to facilitate the transformation and registration of spatial transcriptomic data sets with the anatomical reference space of the Allen Mouse Brain Atlas (Fig. 5). By computationally aligning the spatial transcriptomic data through a digital spot

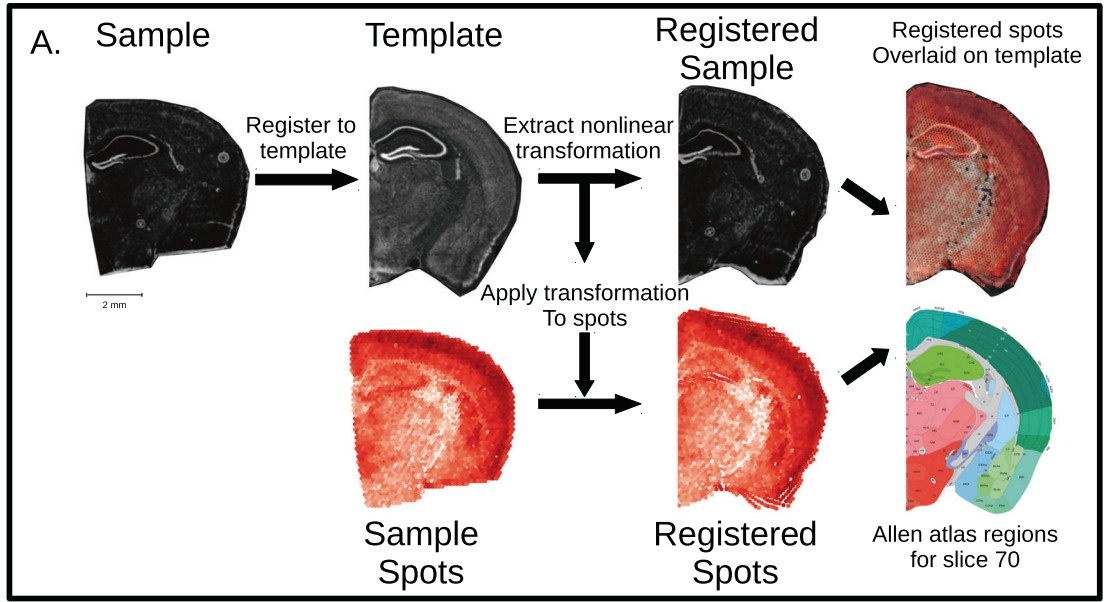

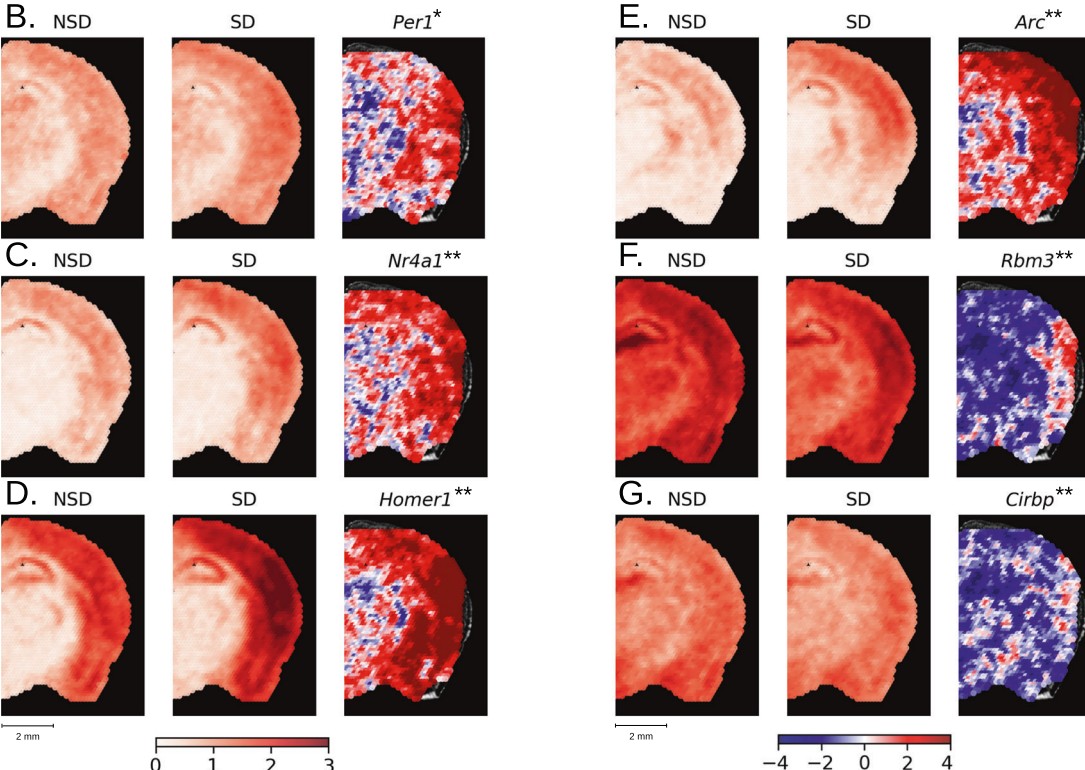

**Fig. 5 | Registration of spatial data to Allen Common Coordinate Framework and statistical analysis of aligned transcriptomic spots. A** Nonlinear registration of the tissue image from a single brain slice ($A_1$) and its transcriptomic spot coordinates ($A_2$)–shown as example: the gene *Camk2n1* – to the template image ($A_3$), slice 70 from the Allen P56 Mouse Common Coordinate Framework (CCF), Allen Mouse Brain Atlas, mouse.brain-map.org. Due to the nonlinear nature of the registration, we were able to precisely align the sample image ($A_4$) to landmarks in the template image and apply that transformation to the spot coordinates ($A_5$). To account for different numbers of spots in individual samples, digital spots spaced at 150 μm in a honeycomb were created for the template slice. Each digital spot is populated with the log base 2 normalized transcriptomic counts from the 7 nearest spots from each sample in a group ($A_7$). This approach allows the comparison of gene expression across entire brain slices in an unrestricted inference space.

**B–G** Samples were split into non-sleep deprived (NSD, *n* = 6, 42 sample spots per digital spot) and sleep deprived (SD, *n* = 7, 49 sample spots per digital spot). The range of the color bar for the mean calculations is set from 0 to a log2 fold-change of 3, the maximum fold change for the genes shown, while the color bar for the SD > NSD t-statistic (**B3–G3**) is bounded to [−4,4], which is approximately the equivalent to the FDR < 0.1. *indicates the gene is significant at FDR < 0.1, **indicates significance at FDR < 0.05. We show a selected group of 6 genes from the 413 DEGs (Supplementary Data 10) (**B–G**). Panel 1 shows for each gene (**B1–G1**) the mean normalized gene count in NSD, panel 2 depicts the mean normalized gene count in SD (**B2–G2**) and panel 3 shows the t-statistics (**B3–G3**). The following DEGs are depicted: **B** *Per1*, 4 significant spots. **C** *Nr4a1*, 29 significant spots. **D** *Homer1*, 306 significant spots. **E** *Arc*, 168 significant spots. **F** *Rbm3*, 31 significant spots. **G** *Cirbp*, 9 significant spots.

workflow with the Common Coordinate Framework, we can observe gene expression changes between the sleep deprived and non-sleep deprived conditions for individual genes of interest. This coordinate approach allows significant changes in gene expression to be visualized and analyzed for individual spots across the brain (Fig. 5 and Supplementary Data 10) in greater detail and with much higher sensitivity for localized changes within larger anatomical structures than the region of interest approach above. We used this approach at its most basic level to examine single gene expression across the brain, finding 413 genes that significantly changed after sleep deprivation. However, our data shows that even genes with robust changes after sleep deprivation display regional differences in expression, which emphasizes that sleep deprivation has localized impacts on gene regulation. With the formidable technological advances that have been made over the past decade, specifically those enabling detailed analysis of gene regulation at multiple levels, one of the greatest challenges facing neuroscientists is the integration and management of complex multimodal data sets. There is a critical need to integrate large data sets for spatial and specific cell type characterization of the mouse brain, as the majority of preclinical research is done using the mouse model. The bioinformatic approach for spatial gene expression analysis across brain regions that we developed for this study helps to meet the challenge of integrating complex data sets for mouse spatial transcriptomic data sets and reveals critical regional selectivity in the impact of brief periods of sleep loss across the brain.

## Methods

### Animals

Male C57BL/6J mice (Jackson Laboratory #000664), age 2.5–3.5 months were used for all the experiments. Mice were group housed (up to 5 per cage) in cages containing soft bedding with food (NIH-31 irradiated modified mouse diet #7913) and water available *ad libitum* in a 12 h :12 h light-dark schedule. The start of the lights-on period is defined as Zeitgeber time zero (ZT 0). Experiments were conducted according to National Institutes of Health guidelines for animal care and use and were approved by the Institutional Animal Care and Use Committee (IACUC) at the University of Iowa.

### Sleep deprivation

All mice were single housed 7 days prior to the experiment with corncob bedding (Envigo, Teklad ¼" corncob, #7907) and soft bedding for nesting. Mice had *ad libitum* access to food and water during sleep deprivation. Mice were habituated for 5 days prior to the experiment by the researcher conducting the experiments. Habituation, performed in the behavior room for experiments, was done by holding each mouse in the palm for 2 min and then after returning to the home cage, tapping the cage for 2 min. Sleep deprivation was performed for 5 h from ZT 0 – ZT 5 using the gentle handling method[32,33]. Briefly, the experimenter tapped the side of the cage, as needed, to keep each mouse awake. When taps were no longer sufficient the mice received a light "cage shake" to rouse the animal. NSD mice remained in the colony housing room throughout the 5 h period.

### Tissue processing and data generation

Each mouse was rapidly euthanized by cervical dislocation at ZT 5 with the whole brain rapidly extracted and flash frozen by ≥ −70 °C isopentane (*n* = 8 SD and *n* = 8 NSD). Frozen brains were stored at −80 °C. Prior to sectioning, a small tissue sample from the cerebellum of each frozen brain was removed, RNA extracted and quality assessed using RNA Integrity Number (RIN). Brains with a RIN above 7 were embedded in optimal cutting temperature medium (OCT) and cryosectioned at −20 °C (10μm sections) with the Leica CM3050 S Cryostat in the Iowa Neuroscience Institute (INI) NeuroBank Core. One coronal section per mouse, corresponding approximately to section 45 of the Paxinos Mouse Brain atlas, was mounted on Visium Spatial Gene Expression

Slides (catalog no. 2000233, 10x Genomics). For tissue collection, we selected a slice that resembled section 45 of the Paxinos Mouse Brain Atlas based on the following features: the shape of the dentate gyrus and fit of CA1 and CA3 bands to the atlas; size and shape of fimbria; size and separation of optic tract and internal capsule; size of lateral ventricle. Sections were immediately processed with the 10x Genomics Visium Gene Expression Slide kit. Full details on the methods used are found in the manufacturer's instructions (CG000239 Rev A User Guide Visium Spatial Gene Expression Reagent Kits). First, the slides were fixed in chilled methanol at −20 °C then stained with hematoxylin and eosin (H&E) to visualize the slices. Brightfield images of the H&E-stained sections were acquired (20X) using an Olympus BX61 Upright Microscope. Raw images were stitched together with the CellSens software (Version 3.2; Olympus) and exported as tiff files. Tissue was then permeabilized with Permeabilization Enzyme (provided by 10X Genomics in the Visium Gene Expression Slide & Reagent Kit, PN-1000184) for 18 min as determined based on tissue optimization time-course experiments. Permeabilization resulted in the release of polyA mRNA from the tissue enabling capture by poly(dT) primers precoated on the Visium Gene Expression slides. Slides also contained barcoded probes with unique molecular identifiers (UMI) so that the spatial gene distribution was mapped. After reverse transcription and second strand synthesis, the amplified cDNA samples from the slides were transferred, purified, and quantified for library preparation. Sequencing libraries were prepared by the Iowa Institute of Human Genetics (IIHG) Genomics Division, according to the Visium Spatial Gene Expression User Guide. Libraries were pooled for sequencing to achieve sequencing depth balance across the samples based on the relative area of coverage of each tissue on the slide. The fragmented cDNA pools were sequenced using an Illumina NovaSeq 6000 SP or S1 flowcell running 100 cycle SBS chemistry v1.5 and aimed for 200 million total read pairs. Read 1 was 48 nucleotide length (10 nt i5 index + 10 nt i7 index + 28 nt Spatial Barcode, UMI) and read 2 was 90 nucleotides length (insert).

### Data processing

Raw FASTQ files and histology images were processed with the Space Ranger software v.1.3.1, which uses STAR v.2.7.10a for genome alignment against the Cell Ranger mm10 reference genome refdata-gex-mm10-2020-A, available at: https://cf.10xgenomics.com/supp/spatial-exp/refdata-gex-mm10-2020-A.tar.gz. QC metrics returned by this software are available in Supplementary Table 1. Quantification and statistical analysis were done with Partek Flow package (Build version 10.0.21.0621) in the Iowa Institute of Human Genetics (IIHG) Genomics Division. Briefly, to avoid raw gene expression counts of 0, a value of 0.001 was added to all counts prior to running SCTransform for normalization and scaling steps. Interpretation of spatial transcriptomic data requires effective preprocessing and normalization to remove spot-to-spot technical variability such as the number of molecules detected in each spot, which can confound biological heterogeneity with technical effects. Recently, a modeling framework for normalization and variance stabilization of molecular count data was made available for spatial datasets, which improves downstream analytical tasks including gene selection, dimensional reduction, and differential expression[68] from spatial datasets. After applying this modeling framework, the dimensionality of each sample was reduced using 100 principal components from the variance of the features. A graph-based clustering was performed to identify the transcriptional signatures of each spot using the Louvain clustering algorithm that includes 30 nearest neighbors and 20 principal components. The Louvain algorithm is an unbiased approach connecting each sample spot to its nearest neighbor. The strength of the connections is weighted based on the similarity between the spots, and higher weight is given to spots more closely related. We then applied the Louvain algorithm to identify "communities" of spots that are more connected to spots in the

same community than they are to spots of different communities, resulting in clustering of the sample spots. The threshold of 20 principal components was chosen based the elbow plot of each sample where most of the transcriptional variation was captured within the first 20 principal components (Supplementary Fig. 8).

The identification of anatomical brain regions for all 16 samples can be found in Supplementary Fig. 2. SCTransform algorithm yields a clearer representation of the different brain regions than a classic log-transformation (Supplementary Fig. 9). However, it is not suitable for differential gene expression analyses, as previously the SCTransform algorithm has been shown to result in ten times more significant false positives when used for differential gene expression analysis[69]. To overcome this challenge, output data from the Space Ranger pipeline were renormalized with the log transformation approach including Counts Per Million (each gene's raw read count in a sample divided by the total number of counts per million in a sample), with a value of 1 added to avoid 0 counts and errors in differential analysis, and finally a log base 2 transformation applied to all values to model and measure proportional fold changes. This normalization revealed similar counts variation across samples (Supplementary Fig. 10). The cluster and brain region labels previously computed by the SCTransform algorithm were then transferred to this log-transformed data. Differential gene expression analysis was performed using the non-parametric Kruskal–Wallis rank sum test because the distribution of the counts does not conform to a normal or binomial distribution (Supplementary Fig. 11). Rank-sum tests have been the most widely used approach in the field of single-cell transcriptomics[70] because it is assumed that every cell (or spot for spatial transcriptomics) is an identical replicate that defines the sample size of the statistics and this approach generates fewer false positives. In this study, the Kruskal–Wallis test was able to assign a median count of 1 (or 0 in log2), for both conditions, for a gene that is not expressed in a given brain region resulting in a fold change of 1 (or 0 in log2) (Supplementary Data 2, differential gene expression analysis in each brain region). Therefore, a gene was considered significantly differentially expressed (DE) if it has a false discovery rate (FDR) step-up (p-value adjusted) below 0.001 and a log2fold-change $\geq |0.2|$.

Addressing concerns regarding the potential for false negatives, we conducted a thorough assessment of the power of our spatial transcriptomic approach. We maintained a robust sequencing depth, with an average of 30,000 UMI counts per spatial spot within the neocortex, consistently across samples and specific neocortical layers such as layer 2/3 (Supplementary Fig. 12). Minimal variability in UMI counts was observed between samples and within the neocortex and neocortical layer 2/3, underscoring the reliability and reproducibility of our methodology (Supplementary Fig. 13). Notably, we identified 12 DEGs common to each neocortical layer after sleep deprivation. The UMI variability of these sleep responsive DEGs did not differ from that of 12 genes unaffected by sleep deprivation in the neocortical layers (Supplementary Fig. 14). This comparative analysis provides strong evidence against false positives, supporting the interpretation that our findings reflect genuine biological changes rather than artifacts.

## Deconvolution: integration with single-cell data
At 55 μm, spots from the Visium platform encompass the expression profiles of 10–20 cells and represent averaged expression of the heterogeneous mixture of cells at the spot level. For this reason, computational techniques called deconvolution have been developed that use scRNA-seq data to infer cell proportions in bulk transcriptomic samples[71]. Consequently, deconvolution of each of the spatial voxels was performed to predict the underlying composition of cell types. We used a reference scRNA-seq dataset of ~14,000 adult mouse cortical cell taxonomy from the Allen Institute[51]. We applied the anchor-based integration that enables the probabilistic transfer of annotations from a reference to a query set, here it is our SCTransformed gene

expression matrix output from Partek Flow®. We then took advantage of the SCTransform normalization to label transfer the cell-type identification of scRNA-seq clusters into the transcriptional signatures of the spatial voxels. The voxels with the highest prediction score were labeled and transferred to the log-transformed data for downstream differential gene expression analysis. The deconvolution of all 16 samples can be found in Supplementary Fig. 5.

## GO molecular function enrichment analyses of differentially expressed genes (DEGs)
The ClueGO[72] and CluePedia[73] plug-ins of the Cytoscape 3.9.0 software[74] were used in "Functional analysis" mode for analyzing the Gene Ontology Molecular Function (4691 terms) database in networks for DEGs. The names of significant DEGs were pasted into the "Load Marker List" of ClueGO, and the organism "*Mus Musculus* [10090]" was selected. Only pathways with a $p < 0.05$ were displayed on the figures (Supplementary Data 4). The GO Term Fusion was used allowing for the fusion of GO parent-child terms based on similar associated genes. The GO Term Connectivity had a kappa score of 0.4. The enrichment was performed using a two-sided hypergeometric test. The p-values were corrected with a Bonferroni step down approach.

## Data and spot preprocessing for STANLY
We inspected all 16 samples visually, excluding any with serious tissue damage or a large amount of tissue folding after adhesion to the slide limiting our analysis to 13 samples. Samples were collected from the left or right hemisphere, but to maximize spatial similarity, we mirrored the right hemisphere samples (2) to the left hemisphere, so that all samples could be aligned in the left hemisphere space. After importing the image data of the slice along with the filtered feature matrix, we reduced the list of spots per slice down to only those listed as "in tissue" by Space Ranger and masked the filtered feature matrix for each sample to first remove empty non-tissue spots. We further removed from the analysis any in tissue spots that had fewer than 5000 total gene counts, which might indicate an error with the spot itself. Any genes that expressed 0 total reads across an entire sample were removed due to low statistical viability. For these 13 samples, the average number of in tissue spots per slide was 2548. Given the localized nature of gene expression to certain tissues or regions of a sample, raw gene counts in each spot are likely to be correlated to their neighbors, but not necessarily across an entire sample. This leads to a high likelihood of a right tail distribution of data when genes are regionally expressed, with potentially high counts in some spots and counts of zero in others. In order to account for this distribution of data, we performed log base 2 normalization on the raw gene counts being fed into the analysis. Log base2 normalization is specifically useful in the case of biological data such as gene counts as this normalizes the data to look for proportional rather than additive changes in expression. STANLY uses 7 neighboring spots in each calculation to account for spatial uncertainty, which is an inevitable problem for every alignment process. Importantly, every spot retains its original log base 2 value, and all 7 spots are utilized in our t-statistics. This strategy reduces spurious findings based on small misalignments.

While this basic principle—the alignment of individual data sets to a common template—is applicable to all kinds of multiomics data sets[75], this version of STANLY has been optimized for the data sets in this study (e.g., regarding resolution, the number of dots, etc.).

## Image preprocessing
Our data was collected as coronal slices of the mouse brain, chosen to be similar to slice 45 in the Paxinos Mouse Atlas, which is similar to Allen Brain Atlas slice 70, so as a template, we chose slice 70 from the Allen Common Coordinate Framework[30]. The code base for image preprocessing steps were performed using SimpleITK[76] (v.5.3.0) and

scikit-image[77] (v.0.19.3) as well as SciPy[78] (v.1.7.3) and NumPy[79] (v.1.21.5) for processing the filtered feature matrices from Space Ranger and performing analysis on the registered spots.

For our current pipeline, most coronal tissue adhered to the slide in such a way that a simple rotation of [0°, 90°, 180, or 270°] is sufficient to bring the tissue images into the same general orientation as the template image. For those images from the right hemisphere, we additionally performed a symmetrical flip on the images and their corresponding spots to match the hemisphere of the template image. This hemisphere combination allows us to maximize the usability of tissue slices in the analysis. Any rotation or mirroring transformation to the tissue image is applied also to the spot coordinates so that these maintain the same space throughout processing. One common problem when trying to register different image modalities is how to handle differences in voxel resolution. In the platform used, we know the size of each spot (55 μm) as well as their distance on center from each other (100 μm). Using the image spot scaling information provided by Space Ranger, we can accurately calculate the size of each spot in the original high-resolution image and calculate the voxel to real world resolution and bring the image into the same resolution as the template. To perform the registration, the tissue image is converted to gray scale. The template image is also min-max normalized to bring it into range of a normal gray scale image rather than the original multi-channel image. To mask the background noise from the sample images we ran a 20 μm Gaussian blur on each image, from which we generated a binary tissue mask using the Otsu method, which allows us to mask out all voxels except for those that contain tissue from the registration process.

## Image registration

After the initial rotation, we selected a single image from our sample set to act as our "best fit." For the best fit, we chose a sample that had good shape and image quality. This selection of a best fit image is done to minimize the need of registering each sample individually to the template image, which has a higher potential for error, and instead register them all to the best fit image that shares more of the image characteristics of H&E stains. To run the registration of the best fit sample (Fig. 5A_1) and its spots (Fig. 5A_2) to the CCF template image (Fig. 5A_3) we used the symmetric image normalization method (SyN) nonlinear registration tools from Advanced Normalization Tools (ANTs)[80] (v.2.3.2), specifically the SyNAggro transformation using a mattes SyN metric with parameters of: SyN sampling = 32, flow sigma = 3, gradient step = 0.1, and registration iterations = [120, 100,80,60,40,20,0]. The result of this registration can be seen applied to the tissue image (Fig. 5A_4) and to the tissue spots (Fig. 5A_5). After the best fit image was registered to the CCF template image, we used the same registration parameters to register the remaining samples to the unregistered best fit image, and then finally applied "best fit to template" transformation generated above to each sample and its spots, bringing them into common space (Fig. 5A_6).

## Digital spots

With all sample images and their spot coordinates in the CCF reference space, we developed a method to create "digital spots" to make running analysis on multiple samples simpler and more closely representative of spacing of the spots in relation to each other. Visium spots are organized in a honeycomb arrangement, where each 55 μm spot has 6 equidistant nearest neighbors spaced 100 μm away on center. Knowing this, we created digital spots that replicate the characteristics of the platform spots in the digital space. Using the 10 μm resolution of the CCF template, we wrote a function that generated a honeycomb spaced grid of digital spots in CCF space and within the bounds of our template mask by defining the desired spacing between digital spots. Due to inevitable spatial uncertainty during registration, we set the spot spacing of our digital sampling to 150 μm in order to "smooth" the

data, a method already common in neuroimaging. We then measured Euclidean distance between each digital spot and template registered tissue coordinates from all samples in the experiment. We sorted these distances and selected at each digital spot from each sample the 7 nearest neighbor spots up to 450 μm, or approximately 3 digital spots away from the center of the digital spot. We chose 7 because of the hexagonal properties of the spot spacing, with every 1 spot having 6 nearest neighbors. Each digital spot is, therefore, a vector of multiple spots from each of the registered samples, e.g., for our 13 samples, this sampling would include up to 7 × 13 sample spots at each digital spot. For our data, this method generated 2052 spots for the CCF template image (Fig. 5A_7), of which we removed 160 spots from analysis for not having sufficient nearest neighbors across samples, leaving 1892 spots. Examples of this sampling can be seen in Fig. 5B–G, with the first image in each plot showing the mean of the digital spots of log base 2 normalized gene counts for NSD samples (Fig. 5B_1–G_1), the second image showing the mean of normalized gene counts for SD (Fig. 5B_2–G_2).

**Statistical analysis of digital spots.** We performed a two-tailed t-test on each digital spot with a Šidák p-value correction (Šidák, 1967) for the number of genes as follows: $\alpha_s = 1 - (1 - \alpha)^{(1/m)}$. Where $\alpha_s$ is the Šidák corrected p-value, α is the original p-value (and m is the number of genes in the transcriptome, n = 18,893). Based on these numbers, any genes that differed between NSD and SD with a p < 2.71e-06 was considered significantly differentially expressed.

The results of the two-tailed t-test for 6 example DEGs can be found in Fig. 5 (Fig. 5B_3–G_3). We used the Šidák method for statistical analysis because it assumes that each test is independent of each other. However, we also tested the Bonferroni and Benjamini–Hochberg methods for FDR correction to verify the strength of the gene analysis. We found the same number of DEGs (413) for the Bonferroni correction as the Šidák, while the Benjamini–Hochberg generated 422 DEGs. The overlap of genes between the Šidák and Bonferroni correction was 100%, with the Benjamini–Hochberg including an additional 9 genes (Supplementary Fig. 15). Thus, spatial transcriptomics provides a robust data set for differential gene analysis irrespective of the method used to correct for multiple comparisons.

**Functional enrichment analysis of DEGs using ToppGene.** ToppFun, the functional enrichment analysis tool from ToppGene suite[52] was run by pasting the list of 413 DEGs generated by STANLY into the ToppFun enrichment gene set and searching for an enrichment of GO: Molecular Functions, GO: Biological Processes, and Mouse Phenotypes (Supplementary Fig. 6).

**RNAscope In situ hybridization.** To validate the results derived from STANLY, we used RNAscope in situ hybridization to assess *Arc* mRNA expression in the hippocampus. RNAscope was performed using commercially available fluorescent reagent kits according to the manufacturer's protocol (Advanced Cell Diagnostics, Inc). In brief, mice were subjected to acute sleep deprivation for 5 h using gentle handling (cage taps and cage shakes). Animals were euthanized by cervical dislocation 1 h after sleep deprivation. Non sleep deprived animals were also euthanized at the same time to eliminate any circadian effect. Brains were dissected from SD and control NSD mice at the same circadian time and fixed for 3 h using 4% paraformaldehyde in PBS at 4 °C. Brains were transferred to 30% sucrose in PBS and kept at 4 °C for 48–72 h. Brains were then cryosectioned at −20 °C (25 μM), with coronal slices placed in a cryoprotective solution (30% sucrose, 30% ethylene glycol in PBS) and stored at −20 °C. Sections were rinsed in cold 1xPBS and then mounted and dried on Superfrost Plus microscope slides (Fisher Scientific, Cat. #12-550-15). Tissue sections on slides were arranged to enable multiple sample conditions on each slide including positive and negative in situ controls, non-sleep deprived sections and sleep deprived sections. For in situ

hybridization, slides were then submerged in 50% ethanol, 70% ethanol, and two 100% ethanol steps for 5 min each at room temperature. Slides were then pretreated with solution according to the manufacturer directions for 30 min, and then washed with 1xPBS twice. The probe RNAscope™ Mm-Arc-C3 (Cat No. 316911-C3) was hybridized to the slides for 2 h at 40 °C. Following hybridization, slides were washed twice with wash buffer at room temperature and then subjected to a series of hybridizations and washes with the AMP 1, AMP 2, AMP 3 and AMP 4 reagents as directed by manufacturer (ACD). Prolong diamond antifade mountant with DAPI (Thermofisher, Cat. # P36962) was used to protect sections and visualize nuclei. Hippocampi were imaged using Leica confocal microscopes.

### Reporting summary

Further information on research design is available in the Nature Portfolio Reporting Summary linked to this article.

## Data availability

The spatial RNA-seq data has been deposited in the National Center for Biotechnology Information (NCBI) Gene Expression Omnibus (GEO) under accession number GSE222410. Data analysis and processing were performed using commercial code from Partek Flow package at https://www.partek.com/partek-flow/. Source data are provided with this paper. Two screenshots of the reference mouse Allen brain atlas (http://atlas.brain-map.org) were used in two figures. The coronal section image 72 was used in Fig. 1C. The coronal section image 70 from the Allen P56 Mouse Common Coordinate Framework (CCF) was used in Fig. 5A3 as well as a template for the alignment step in STANLY. A reference scRNA-seq dataset of ~14,000 adult mouse cortical cell taxonomy from the Allen Institute was used for deconvolution (DOI: 10.1038/nn.4216). The mm10 reference genome used for the alignment of the reads in spatial transcriptomic can be found here: https://cf.10xgenomics.com/supp/spatial-exp/refdata-gex-mm10-2020-A.tar.gz. Source data are provided with this paper.

## Code availability

The code for the deconvolution analysis can be accessed through GitHub (https://github.com/YannVRB/Sleep-deprivation-spatial-transcriptomic.git). The code for STANLY and subsequent analysis can be accessed through GitLab (https://research-git.uiowa.edu/zjpeters/STANLY).

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

## Acknowledgements

The study was supported by the National Institutes of Health R01 Grant (R01AG062398) to T.A. and L.C.Ly., the University of Iowa Hawkeye Intellectual and Developmental Disability Research Center (P50 HD 103556; L. Strathearn and T.A., multi-PIs), and the Hensing Brain and Behavior Fund as well as the Carver Trust. T.N.-J. was supported by the Andrew H. Woods Professorship. T.A. is the Roy J. Carver Chair of Neuroscience. The authors acknowledge Xiaowen Wang's exceptional technical support from Partek Inc., which was crucial for Visium spatial RNAseq data analysis. Data presented herein were obtained at the Iowa NeuroBank Core in the Iowa Neuroscience Institute, and the Genomics Division in the Iowa Institute of Human Genetics, which is supported, in part, by the University of Iowa Carver College of Medicine.

## Author contributions

T.A. designed the study with input from S.C., L.C.Ly. and T.N-J. E.N.W. performed the sleep deprivation and tissue collection. L.-C.Li. performed the tissue preparation for spatial transcriptomics. Y.V. and Z.P. performed the transcriptomic and statistical analysis with the advice and guidance of the senior authors. Z.P. and T.N-J. developed and wrote the code for the STANLY analysis. L.C.Ly. and E.N.W. performed the RNAscope, confocal microscopy and data analysis. Y.V., Z.P., L.C.Ly., and E.N.W. wrote the manuscript with input from all the authors.

## Competing interests

T.A. serves on the Scientific Advisory Board of EmbarkNeuro and is a scientific advisor to Aditum Bio and Radius Health. The other authors declare no conflicting interests.

## Ethics approval

All procedures on mice in this study were approved by the Institutional Care and Use Committee at the University of Iowa.
