## [Peer Review File · Nature Communications]

Spatial transcriptomics reveals unique gene expression changes in different brain regions after sleep deprivation.REVIEWER COMMENTS

Reviewer #1 (Remarks to the Author):

- What are the noteworthy results?

The authors use spatial transcriptomics to study the effects of 5 hours sleep deprivation throughout the murine brain. As previously shown, major changes are found across brain regions, particularly hippocampus, neocortex, hypothalamus and thalamus. These transcriptional changes seem to be region specific and don't necessarily follow a similar pattern as illustrated by major up-regulation of neocortex transcripts but down-regulation of hippocampal transcripts.

Using available single cell RNA-seq data, the authors further deconvoluted hippocampal and cortex spatial information, demonstrating that the effects of sleep deprivation is heterogeneous even within different layers of these regions. Most striking is the evidence found in favor of high number of unique effects of sleep deprivation even between layers of neurons, suggesting highly specific responses based on individual neuronal identities.

The authors further developed a method to register all images to a unified reference space, allowing statistically analyses all images in a comparable manner. This method improves the spatial resolution of the data for the analysis of brain spatial transcriptomics when compared to deconvolution.

- Will the work be of significance to the field and related fields? How does it compare to the established literature? If the work is not original, please provide relevant references.

Given the complexity and heterogeneity of the brain tissues, the description of the different spatial effects of sleep deprivation provides an important new layer of information to understand the impact and function of sleep. To my knowledge, despite the various brain regions having been previously profiled independently, the approach taken in this work is original in that it takes a simultaneous look at the entirety of the brain.

Even though the work clearly focuses on the effects of lack of sleep, the methods described can in principle be applied to any other question, particularly related to brain biology, so the work reported can be significant to neurosciences in general.

However, I feel the authors could comment more on of how do the number of differentially expressed genes compare to the previous reports. Considering the cited works, deep RNA-seq suggests twice as many hippocampal DEG as reported here, with a more balanced distribution between up- and down-regulation, while micro-arrays find similar results. In the authors' opinion, how do these different methods lead to these discrepancies? Would the sensitivity of spacial transcriptomics described in this work be lower compared to bulk RNA-seq methods? Similarly, up to 70% of cortex transcriptome has

been suggested to be altered by sleep deprivation, which is a larger proportion than the one described. Do the authors think these differences can be explained by differences in the sleep deprivation protocol, experimental method or statistical analysis?

- Does the work support the conclusions and claims, or is additional evidence needed?

The conclusions are supported by the results shown. In addition, most of the conclusions are additionally supported by previous research, confirming the ability of spatially transcriptomics to, at least partially, produce similar results to alternative approaches.

If possible, I think alternative demonstration of some of these results by classical molecular methods would be relevant to further support the conclusions. For example, can some of the results of figure 5 be replicated with in situ hybridization or other similar method to validate the observed spatial heterogeneity?

- Are there any flaws in the data analysis, interpretation and conclusions? Do these prohibit publication or require revision?

STANLY makes use of advanced image processing methods and Advanced Normalization Tools (ANTs) and The Allen Software Development Kit (SDK), with which I have no professional experience. Although the described method looks correct and valid from a statistical and computational perspective, I am not an expert in these methodologies.

Regardless of this limitation, I find the authors' arguments in favor of their analysis, interpretation and conclusions well founded. However, since this is a new method, discussing the limitations of this method, both practical (e.g. can it work with all types of data or only Visium data?) and technical (e.g. is there loss of resolution from the averaging of 7 neighbor spots?) in more detail is important to clarify how it can be used for other research questions and by other groups.

I find that the reason why SCTransform is not suitable for differential analysis, but it is suitable for other analyses, is not clear to all readers. Could the authors develop a bit more on this methodological aspect?

- Is the methodology sound? Does the work meet the expected standards in your field?

As above, I believe the work meets expected standards and the methodology is sound, providing a clear example of how emerging technologies can be used to address standing questions in the field.

- Is there enough detail provided in the methods for the work to be reproduced?

Given the time available, I did not re-run the analysis myself. However, all the links provided lead to the respective code or data, so this work should, in principle, be reproducible.

As a comment, STANLY howto.md should be made more visible (e.g. mentioned or merged with readme.md) so potential users can easily find the instructions on how to use the method.

- Other comments

The manuscript often refers to a 10x Genomics Visium platform (starting in the highlights!) Although it would not be unusual to be referred to in the methods section, the way it is frequently mentioned sometimes gives the manuscript the feeling of “advertising the method validation”. Given the commercial nature of this central method to the work, I leave for the editors whether this could be considered a competing interest at some level or just an authors’ oversight.

There’s some minor grammar mistakes that should be corrected during the revision (e.g. "Loss of sleep affects impacts cognition").

STANLY uses hemisphere mirroring to maximize the usability of the tissue slides. Though it is not expected a priori, it would be relevant to demonstrate there are no major asymmetries in the brain response to sleep deprivation.

The enrichment analyses (and the manuscript in general) focus on the unique differential expressed genes for each region/layer. However, exploring if there are any common mechanisms shared between the different parts of the brain could also be of interest. Could the authors comment if any, and which, molecular mechanisms are enriched for shared genes?

- Figures and tables

From supplementary table 1, it looks like the median number of genes per spot is 5978.25 instead of the *mean number* as mentioned in the results section. Additionally, it would seem that for each sample, > 21k genes are detected, which I find highly relevant, as it illustrates the ability to identify most genes, but this is not mentioned in the results section. Furthermore, adding a small comment on the disparity between genes per spot and genes per sample could be relevant to properly interpret these results (is this a regional bias? cell type specific biology? a limitation of the “spot” approach? etc.)

Figure 2D has a green colored gene (Cnp), which is an unexpected color not described in the figure legend.

The supplementary Tables it is unclear why some regions seem to have been filtered, while others contain non-significant genes (e.g. Table S2, allocortex 19327 genes, neocortex 3689). Gene ID also matches gene name, which is unexpected. It should either be a numerical value or the official symbol (eg. Arntl instead of Bmal1). Finally, there’s a number of highly significant genes showing a fold-change

of 1, which I would intuitively think as showing no difference. Could these tables be double checked?

Table S6 is particularly full of NaN values. With the lack of headers, this makes it extremely difficult to interpret.

Figure 5 could benefit having the color code be explicitly defined in the legend (blue down-regulated and red up-regulated) and which t-statistic is represented on the third panel (SD/NSD or NSD/SD)

Deconvolution methods mention that all 16 are showed in figure S4. However, this doesn't seem to be the correct image. Also, there are two supplementary figures S5.

Reviewer #2 (Remarks to the Author):

Vanrobeys et al., in this study report region specific alteration in brain transcriptome in response to sleep deprivation (SD). Their study establishes hippocampus (Hip) and neocortex (NC) as the predominant regions that show alteration in regulation of gene expression in response to SD. Surprisingly, SD drive a heterogeneous response across distinct brain regions. The unique approach of spatial transcriptomics the authors employ also reveals the alteration in the hypothalamus and thalamus in addition to Hip and NC as a consequence of SD. The body of work provides a helpful approach for the analysis of brain-section wide alterations in transcriptome and integrate it to specific anatomical regions for visualization. In addition to this the method developed by the authors will assist in identifying transcriptomic alterations, associating them to very small brain regions inaccessible with traditional methods.

The study is well conducted, discovers unique patterns of transcriptomic changes in multiple brain regions and recapitulates previous discoveries. Thus, demonstrating the sensitivity of the approach and its ability to make new discoveries and visualize them in an anatomical space. Another strength of this method is that it utilizes existing brain atlases to look at subregion-specific alterations in transcriptomics which can contribute to our understanding of region-specific responses. The development of STANLY a spatial transcriptome tool is a necessary advancement and growth in new direction for spatial transcriptomics in brain biology likely to be of great value to many researchers. The figures are excellently prepared, and the conclusions are well supported by the presented data. There are some technical clarifications and points the authors need to address to further enhance the clarity of result interpretation and better substantiate the conclusions.

Comments:

1. In figure 1D, the UMAP clusters the Allocortex and striatum like amygdalar nuclei in to two clusters. Since the same region is clustered twice, could the authors provide an explanation if this is correct or check the data to correct for this.

2. In the result section line 100 and 101 the authors mention that unsupervised clustering was used with

transcriptional signatures, the authors should mention which of the unsupervised clustering methods was used.

3. In the figure legend for fig 3 line 813, the authors seem to have used an FDR step up <0.1 while in other figures, the FDR step up is reported as <0.001 . Why was a different metric used for fig 3 specifically? This appears to be a typing error. Please elaborate.

4. For the statistical analysis of digital spots (Fig. 5), a Sidak p-value correction was used. Could the authors provide a rationale for the selection of Sidak's p-value correction in the methods section. The Sidak method enhances the statistical power thus increasing the probability of false positives in comparison to more advanced/conservative methods of p-value correction. In this case, the Sidak correction with the criterion that at least 3 digital spots of the 1892 might still contribute to false positives despite the new critical value after the correction. It remains unclear how the selection of 3 spots was arrived at. Therefore, we recommend that other more conservative p-value correction methods be used to control for false discoveries.

5. Additionally, in fig 5(B1-G3) the spot comparison reduces the spatial resolution of its location, would it be possible within the scope of the project to integrate/super-impose a line map of the brain region for spatial reference.

6. In the methods section, the authors specify that a coronal section corresponding/similar to the 45th slice in the Paxinos mouse atlas. Could the authors specify which subregions this corresponds to since the brain regions mentioned can refer to a broader anatomical region.

Reviewer #3 (Remarks to the Author):

The study titled, "Spatial transcriptomics reveals unique gene expression changes in different brain regions after sleep deprivation", concerns a study employing newly developed spatial transcriptomic techniques and analytical methods to examine spatially localized DEGs in a coronal section of mouse brain that includes hippocampus, neocortex and subcortical regions, in response to acute sleep deprivation. The findings extend previous bulk tissue transcriptomic studies by allowing comparison of different brain regions' DEG from 13 samples (the number of samples analyzed for different aspects of the study is not always made clear). The MS is clearly written, nicely illustrated, and the study is well designed and appears to be technically sound.

The expression patterns derived from the spots' transcriptomes correspond to anatomically defined brain regions. By registering $\sim 50\mu\text{m}$ diameter spots' DEGs across the samples comprising each sleep group to a common anatomical reference location, the authors identify sets of DEGs specific to localized regions. One of the most interesting findings is that most hippocampal DEGs are downregulated and the majority of neocortical DEGs are upregulated. Also, this is a technically novel application of spatial transcriptomic analysis to changes induced by sleep deprivation.

DEGs unique to localized brain regions (hippocampal sub-regions or neocortical layers) are also

identified. Although FDR appears to be well controlled, the power of the spatial transcriptomic technique for DEGs may be problematic. According to the authors, the spots may involve 10-20 different cells so that a spot may better resemble highly localized bulk tissue in that a particular gene in a given spot may be up or downregulated or change very little and this might vary considerably from spot to spot depending on the cellular makeup of the spot (for example no DEGs were identified in CA2 and CA3). Further, even with scRNAseq, using a negative binomial mixed model like NEBULA, power, estimated from realistic simulations is less than 50% for several thousands of single cells of a given cell type (many more cells than spots, each spot with mixed cells) to detect DEGs with a FC of ~ 1.2 . Is it possible that at least some of the anatomically unique cells might be identified due to a very high rate of false negatives? Perhaps this could be more directly addressed in the discussion and the conclusions revised accordingly.

REVIEWER COMMENTS

Reviewer #1 (Remarks to the Author):

Comment 1:

- What are the noteworthy results?

The authors use spatial transcriptomics to study the effects of 5 hours sleep deprivation throughout the murine brain. As previously shown, major changes are found across brain regions, particularly hippocampus, neocortex, hypothalamus and thalamus. These transcriptional changes seem to be region specific and don't necessarily follow a similar pattern as illustrated by major up-regulation of neocortex transcripts but down-regulation of hippocampal transcripts.

Using available single cell RNA-seq data, the authors further deconvoluted hippocampal and cortex spatial information, demonstrating that the effects of sleep deprivation is heterogeneous even within different layers of these regions. Most striking is the evidence found in favor of high number of unique effects of sleep deprivation even between layers of neurons, suggesting highly specific responses based on individual neuronal identities.

The authors further developed a method to register all images to a unified reference space, allowing statistically analyses all images in a comparable manner. This method improves the spatial resolution of the data for the analysis of brain spatial transcriptomics when compared to deconvolution.

Response: We thank the reviewer for the supportive comments pointing out our noteworthy findings.

Comment 2: Will the work be of significance to the field and related fields? How does it compare to the established literature? If the work is not original, please provide relevant references.

Given the complexity and heterogeneity of the brain tissues, the description of the different spatial effects of sleep deprivation provides an important new layer of information to understand the impact and function of sleep. To my knowledge, despite the various brain regions having been previously profiled independently, the approach taken in this work is original in that it takes a simultaneous look at the entirety of the brain.

Even though the work clearly focuses on the effects of lack of sleep, the methods described can in principle be applied to any other question, particularly related to brain biology, so the work reported can be significant to neurosciences in general.

Response: We thank the reviewer for this kind assessment of the originality and significance of our work.

Comment 3: However, I feel the authors could comment more on of how do the number of differentially expressed genes compare to the previous reports. Considering the cited works, deep RNA-seq suggests twice as many hippocampal DEG as reported here, with a more balanced distribution between up- and down-regulation, while micro-arrays find similar results. In the authors' opinion, how do these different methods lead to these discrepancies? Would the

sensitivity of spatial transcriptomics described in this work be lower compared to bulk RNA-seq methods? Similarly, up to 70% of cortex transcriptome has been suggested to be altered by sleep deprivation, which is a larger proportion than the one described. Do the authors think these differences can be explained by differences in the sleep deprivation protocol, experimental method or statistical analysis?

Response: We appreciate the reviewer's questions. We do not think the differences arise from the sleep deprivation protocol, as the protocol is similar to what was used in previous studies. We also do not believe the differences arise from the experimental methods used in spatial transcriptomics, as we find a similar result in our subregional analysis of genes such as *Arc* compared to what has been reported in other studies. As analysis of spatial transcriptomic data is new in the field, we have adopted a highly conservative approach for data analysis. With our conservative approach, we do find a lower number of significant differentially expressed genes compared to previous transcriptomic studies which used RNA sequencing or microarrays. Given the large number of data points per sample in spatial transcriptomic analysis, encompassing thousands of spots, the initial adoption of a highly conservative approach with a very low FDR is necessary because the large sample size may increase the number of outliers or false positives. Consequently, we used a rank-sum Kruskal-Wallis test for our statistical analysis. However, as with many RNA-seq transcriptomic studies, the FDR can be set at different thresholds to identify differentially expressed genes.

We have included new text in the Discussion of the revised manuscript to address these questions.

New text, Discussion p. 12:

Using a conservative FDR-corrected p-value of <0.001 as the threshold, we identified fewer differentially expressed genes in the hippocampus and the cortex compared to other transcriptomic studies^{11,13,21}. As our sleep deprivation method and protocols were similar to other studies^{11,13}, we believe that differences arise from our use of a highly conservative statistical approach to avoid false positives due to the large sample size (i.e., the number of spots in each slice). However, we recognize that our conservative approach may also result in false negatives, so all gene expression data is included in the Supplemental Tables and available through GEO (GSE222410).

Comment 4:

- Does the work support the conclusions and claims, or is additional evidence needed?

The conclusions are supported by the results shown. In addition, most of the conclusions are additionally supported by previous research, confirming the ability of spatially transcriptomics to, at least partially, produce similar results to alternative approaches.

If possible, I think alternative demonstration of some of these results by classical molecular methods would be relevant to further support the conclusions. For example, can some of the results of figure 5 be replicated with *in situ* hybridization or other similar method to validate the observed spatial heterogeneity?

Response: As suggested by the reviewer, we have included new experimental data using RNAscope (Supplemental Figure 7), an *in situ* hybridization method, to analyze the

expression pattern of *Arc* mRNA in the hippocampus. We then compared these results to subregional analysis of spatial transcriptomic data using STANLY. Both methods yield similar findings, with sleep deprivation increasing *Arc* mRNA expression in area CA1 but not in the dentate gyrus. Thus, the results from the RNAscope analysis for the spatial heterogeneity of *Arc* gene expression in the hippocampus after sleep deprivation validate subregional analysis using our new computational approach, STANLY.

New Text added to the revised manuscript, Methods p. 28:

RNAscope *In situ* hybridization

To validate the results derived from STANLY, we used RNAscope *in situ* hybridization to assess *Arc* mRNA expression in the hippocampus. RNAscope was performed using commercially available fluorescent reagent kits according to the manufacturer's protocol (Advanced Cell Diagnostics, Inc). In brief, mice were subjected to acute sleep deprivation for 5 hours using gentle handling (cage taps and cage shakes). Brains were dissected from SD and control NSD mice at the same circadian time and fixed for three hours using 4% paraformaldehyde in PBS at 4° C. Brains were transferred to 30% sucrose in PBS and kept at 4° C for 48 – 72 hours. Brains were then cryosectioned at -20 °C (25 μM), with coronal slices placed in a cryoprotective solution (30% sucrose, 30% ethylene glycol in PBS) and stored at -20° C. Sections were rinsed in cold 1xPBS and then mounted and dried on Superfrost Plus microscope slides (Fisher Scientific, Cat. #12-550-15). Tissue sections on slides were arranged to enable multiple sample conditions on each slide including positive and negative *in situ* controls, non-sleep deprived sections and sleep deprived sections. For *in situ* hybridization, slides were then submerged in 50% ethanol, 70% ethanol, and two 100% ethanol steps for 5 min each at room temperature. Slides were then pretreated with solution according to the manufacturer directions for 30 min, and then washed with 1xPBS twice. Probes were hybridized to the slides for 2 hours at 40° C. Following hybridization, slides were washed twice with wash buffer at room temperature and then subjected to a series of hybridizations and washes with the AMP 1, AMP 2, AMP 3 and AMP 4 reagents as directed by manufacturer (ACD). Prolong diamond antifade mountant with DAPI (Thermofisher, Cat. # P36962) was used to protect sections and visualize nuclei. Hippocampi were imaged using a Leica confocal microscope.

New text added to the revised manuscript. Results section p. 11:

We wanted to test the power of STANLY and spatial transcriptomics for subregional analysis within the hippocampus, particularly area CA1 and the dentate gyrus as regions-of-interest for this validation. Both area CA1 and the dentate gyrus were treated as binary masks, with dots situated within these regions included into our analyses. We then conducted a two-sample t-test at each spot within these masks as shown in Supplemental Figure 7. Because we were only investigating one gene, *Arc*, we performed our p-value correction using the number of spots being tested, which was 140, giving us a threshold of $p < 0.00075$. Using this p-value, we found that sleep deprivation significantly increased the mRNA expression of the activity-dependent immediate early gene *Arc* in area CA1 ($p < 0.0005$), whereas there was little change or even slightly decreased expression of *Arc* in the dentate gyrus. To validate the reliability of our STANLY analyses, using samples from independent sleep deprivation experiments, we performed *in situ* hybridization using RNAscope (ACD) for *Arc* expression in the hippocampus. The RNAscope analysis revealed that acute sleep deprivation significantly increased *Arc* mRNA positive cells in the CA1, while there was no change in the DG (Sup. Fig. 7). *Arc* shows increased expression in the hippocampus following acute sleep deprivation^{11,13}. Moreover, *Arc* has also been identified as having differential expression in the subregions of the hippocampus after sleep deprivation with significantly increased levels of *Arc* in the CA1 and no change or

decreased expression in the dentate gyrus⁵². Thus, the results from STANLY analysis comparing subregions within the hippocampus are consistent with our *in situ* hybridization experiments and with previously published results. These results validate the power of STANLY and spatial transcriptomic approaches to identify spatially restricted changes in gene expression.

Additional Supplemental Figure 7

Comment 5:

• Are there any flaws in the data analysis, interpretation and conclusions? Do these prohibit publication or require revision?

STANLY makes use of advanced image processing methods and Advanced Normalization Tools (ANTs) and The Allen Software Development Kit (SDK), with which I have no professional experience. Although the described method looks correct and valid from a statistical and computational perspective, I am not an expert in these methodologies.

Regardless of this limitation, I find the authors arguments in favor of their analysis, interpretation and conclusions well founded.

Response: We thank the reviewer for their support of our data analysis, interpretations, and conclusions.

Comment 6:

However, since this is a new method, discussing the limitations of this method, both practical (e.g. can it work with all types of data or only Visium data?) and technical (e.g. is there loss of resolution from the averaging of 7 neighbors spots?) in more detail is important to clarify how it can be used for other research questions and by other groups.

Response: We thank the reviewer for giving us the opportunity to further elaborate on these important issues. The basic principle – the alignment of the spots of individual data sets to a common template – is applicable to a broad range of spatial transcriptomics data sets. However, we have optimized this method for the data set that we used in this study regarding resolution, the number of dots, etc.

We have added new text in the Methods, p. 24:

While this basic principle – the alignment of individual data sets to a common template – is applicable to all kinds of multiomics data sets⁷⁴, this version of STANLY has been optimized for the data sets in this study (e.g., regarding resolution, the number of dots, etc.).

Response continued: The use of 7 neighboring spots is intended to account for spatial uncertainty that is inevitable due to the alignment process. Each spot retains its original log base 2 value, and all 7 spots are used in the calculation of the t-statistic. It is a strategy to counter spurious findings based on misalignments. This should reduce the amount of false positive results that are not reflective of the “true” differences between groups but rather driven by technical artifacts. This strategy is very common in neuroimaging, where it is referred to as “smoothing.” Although this strategy reduces spurious findings based on spatial uncertainty, it also slightly reduces the resolution. We’ve added the following sentence to the manuscript to elaborate on this:

New text in the Methods section, p. 24:

STANLY uses 7 neighboring spots in each calculation to account for spatial uncertainty, which is an inevitable problem for every alignment process. Importantly, every spot retains its original log base 2 value, and all 7 spots are utilized in our t-statistics. This strategy reduces spurious findings based on small misalignments.

Comment 7:

I find that the reason why SCTransform is not suitable for differential analysis, but it is suitable for other analyses, is not clear to all readers. Could the authors develop a bit more on this methodological aspect?

Response: The SCTransform algorithm yields a more accurate representation of different brain regions, as pictured below, enabling distinction between the neocortex and the allocortex, clear separation of the spots in the hippocampal region and the neocortex and differentiation between the subregions of the striatum. However, the authors of the SCTransform algorithm demonstrated that differential expression using SCTransform resulted in ten times more significantly differentially false-positive genes (Choudary and Satija, 2022 PMID: 35042561). Thus, it is necessary to use a different analysis approach for differential gene expression. By using both SCTransformed and logtransformed data we can better identify brain regions (see figure below) and more appropriately analyze differential gene expression. We have now more fully explained this in the revised manuscript.

New text in the Methods, p. 21. Line 452

The identification of anatomical brain regions for all 16 samples can be found in **Supplemental Figure 2**. SCTransform algorithm yields a clearer representation of the different brain regions than a classic log-transformation (**Sup. Fig. 9**). However, it is not suitable for differential gene expression analyses, as previously the SCTransform algorithm has been shown to result in ten times more significant false positives when used for differential gene expression analysis⁶⁸. To overcome this challenge, output data from the Space Ranger pipeline were renormalized with the log transformation approach including Counts Per Million (each gene's raw read count in a sample divided by the total number of counts per million in a sample), with a value of 1 added to avoid 0 counts and errors in differential analysis, and finally a log base 2 transformation applied to all values to model and measure proportional fold changes.

Comment 8:

- Is the methodology sound? Does the work meet the expected standards in your field?

As above, I believe the work meets expected standards and the methodology is sound, providing a clear example of how emerging technologies can be used to address standing questions in the field.

- Is there enough detail provided in the methods for the work to be reproduced?

Given the time available, I did not re-run the analysis myself. However, all the links provided lead to the respective code or data, so this work should, in principle, be reproducible.

As a comment, STANLY howto.md should be made more visible (e.g. mentioned or merged with readme.md) so potential users can easily find the instructions on how to use the method.

Response: We would like to thank the reviewer for pointing out this issue. We have added a direct link to the howto page in the readme which is displayed on the homepage of the git repository.

Comment 9:

The manuscript often refers to a 10x Genomics Visium platform (starting in the highlights!) Although it would not be unusual to be referred to in the methods section, the way it is frequently mentioned sometimes gives the manuscript the feeling of “advertising the method validation”. Given the commercial nature of this central method to the work, I leave for the editors whether this could be considered a competing interest at some level or just an authors’ oversight.

Response: The 10x Genomics Visium platform is a new technology and could be compared with other established sequencing methods, therefore we emphasized the name of the technique and the platform we used. This was an oversight on our part and we did not realize it would come across this way. We have revised the manuscript to avoid the overuse of the phrase “10X Genomics Visium platform” or “Visium” to preclude leaving this impression on readers.

Comment 10:

There’s some minor grammar mistakes that should be corrected during the revision (e.g. "Loss of sleep affects impacts cognition").

Response: We thank the reviewer for bringing this to our attention. We have looked closely over the manuscript to identify and correct any grammar mistakes.

Comment 11:

STANLY uses hemisphere mirroring to maximize the usability of the tissue slides. Though it is not expected a priori, it would be relevant to demonstrate there are no major asymmetries in the brain response to sleep deprivation.

Response: Due to the size limitations of spatial transcriptomic slides, it is not possible to fit an entire coronal brain slice on the slide, so only one hemisphere of the brain section was used for each animal. Almost all of our samples were from the right hemisphere (5 for SD, 6 for NSD) with only one left hemisphere section for SD and one for NSD used in the STANLY analysis. Thus, we do not have enough replicates to make a comparison between changes in gene expression after sleep deprivation in the left and right hemispheres. We have re-run our analysis using just the right hemisphere samples and find similar results for the number of differentially expressed genes. In future studies, we plan to investigate potential differences in the impact of sleep deprivation between the left and right brain hemispheres.

We have included new text in the Discussion, p. 13:

In the current study, we were unable to investigate potential asymmetries in the response of the left and right hemisphere to the impacts of acute sleep deprivation, as we were limited by the size of the spatial transcriptomic slides in which only one hemisphere can fit within the bounds

of the slide. In future studies, we plan to use spatial transcriptomics to investigate more nuanced impacts of sleep deprivation, such as interhemispheric differences.

Comment 12:

The enrichment analyses (and the manuscript in general) focus on the unique differential expressed genes for each region/layer. However, exploring if there are any common mechanisms shared between the different parts of the brain could also be of interest. Could the authors comment if any, and which, molecular mechanisms are enriched for shared genes?

Response: We also questioned whether there were conserved effects of sleep deprivation across brain regions. To our surprise, there was very little overlap in the genes that were affected by sleep deprivation between brain regions. Strikingly, we only found two regions, the neocortex and the allocortex, that shared a great enough number of DEGs to perform pathway analysis. There were 35 DEGs in common between the neocortex and the allocortex with molecular function enrichment analysis highlighting protein kinase inhibitor activity. All other sets of common DEGs were too small to reveal enrich significant molecular functions. In the revised manuscript, we have created a supplemental figure to illustrate the molecular function for shared genes between the Neocortex and the Allocortex (Supplemental Figure 4). Differentially expressed genes common between brain regions are listed in Supplemental Table 5. We have revised the text to reflect the inclusion of this information.

Neocortex and allocortex

New text in Results, page 7:

Of the 592 DEGs found in the hippocampal region, 489 were exclusively affected in the hippocampal region (489/592 DEGs), 306/401 in the neocortex, 199/266 in the hypothalamus, 56/113 in the thalamus, and 33/66 in the striatum-like amygdalar nuclei (**Sup. Table 3**). Interestingly, only 35 DEGs were found to be in common between the neocortex and the allocortex, resulting in one significantly enriched pathway: protein kinase inhibitor activity (**Sup. Fig. 4**). All other sets of common DEGs (**Sup. Table 5**) were too few in number to reveal enriched molecular functions.

Comment 13:

- Figures and tables

From supplementary table 1, it looks like the median number of genes per spot is 5978.25 instead of the *mean number* as mentioned in the results section. Additionally, it would seem that for each sample, > 21k genes are detected, which I find highly relevant, as it illustrates the

ability to identify most genes, but this is not mentioned in the results section. Furthermore, adding a small comment on the disparity between genes per spot and genes per sample could be relevant to properly interpret these results (is this a regional bias? cell type specific biology? a limitation of the “spot” approach? etc.)

Response: We apologize for the confusion about the number of genes and the labels. In Supplementary Table 1, the median number of genes per spot can be found for each sample. In the revised table, we have listed the mean value of the medians across the samples to provide the mean number of genes per spot. The value of 5978.25 genes per spot refers to the mean of the median of genes per spot within each sample. The number of genes per spot is indeed 3 - 4 times lower than the number of genes per sample. This is primarily due to cell-type and brain-region specific gene expression such that only a subset of genes is expressed in each spot. This is illustrated in the figure below where an example is displayed for the number of expressed genes across the tissue section. The highest number of genes can be found in excitatory neurons residing in the hippocampal region and cortex. We have added this figure as a Supplemental Figure (Supplemental figure 1) and expanded the result section to explain this.

New text in the Results, page 5:

Importantly, we were able to detect over 21,000 genes for each sample (Sup. Table 1). However, the individual number of genes detected in each spot is three to four times lower than the total number of expressed genes detected due to cell-type and brain-region specific differences in gene expression across brain regions (Sup. Fig 1).

Comment 14:

Figure 2D has a green colored gene (Cnp), which is an unexpected color not described in the figure legend.

Response: We apologize for this mistake, and we thank the reviewer for pointing it out. The color code for the Cnp genes has been changed from green to red.

Comment 15:

The supplementary Tables it is unclear why some regions seem to have been filtered, while others contain non-significant genes (e.g. Table S2, allocortex 19327 genes, neocortex 3689). Gene ID also matches gene name, which is unexpected. It should either be a numerical value or the official symbol (eg. Arntl instead of Bmal1). Finally, there's a number of highly significant genes showing a fold-change of 1, which I would intuitively think as showing no difference. Could these tables be double checked?

Response: We apologize for the confusion in Supplemental Table 2 and the mistake with the Gene ID. In the analysis, significant DEGs were identified using two filters, an FDR of 0.001 and an absolute fold change threshold of 1.2. Table 2 originally showed genes that

fit either of those criteria, instead of only fitting both. Now, in Supplemental Table 2, the expression for all genes is displayed with no filters and all the data is available. We have also now included a new Table (Supplemental Table 3) showing only the significant differentially expressed genes. We have revised the Results to more clearly explain the filters and the designation of significant DEGs.

Revised Text in the Results, p. 5:

Supplemental Table 2 reports the expression levels for all genes with no filters.

Revised Text in the Results, p. 6:

In the analysis, significant DEGs were identified using two filters, an FDR of 0.001 and an absolute fold change threshold of 1.2. Significant DEGs for each brain region are reported in Supplemental Table 3.

Comment 16:

Table S6 is particularly full of NaN values. With the lack of headers, this makes it extremely difficult to interpret.

Response: We removed NaN values and have added headers representing the coordinate location of the spot.

Comment 17:

Figure 5 could benefit having the color code be explicitly defined in the legend (blue down-regulated and red up-regulated) and which t-statistic is represented on the third panel (SD/NSD or NSD/SD).

Response: Heatmaps shown in B-G are color coded based on the result of the SD/NSD t-statistic of the log base 2 normalized data, with blue meaning a decrease in expression in SD animals, and red signifying an increase in SD animals. White spots indicate where the mean is the same between the two groups. We have updated the figure and figure legend to clarify. We have also added asterisks to distinguish between significance levels.

For the revised version of the manuscript, we have now added scales to the color codes and the following changes to the figure description for Figure 5 (page 43):

B-G. Samples were split into non-sleep deprived (NSD, n=6, 42 sample spots per digital spot) and sleep deprived (SD, n=7, 49 sample spots per digital spot). The range of the color bar for the mean calculations is set from 0 to a log2 fold-change of 3, the maximum fold change for the genes shown, while the color bar for the SD > NSD t-statistic (B3-G3) is bounded to [-4,4], which is approximately the equivalent to the FDR < 0.1. * indicates the gene is significant at FDR < 0.1, ** indicates significance at FDR < 0.05. We show a selected group of 6 genes from the 413 DEGs (Sup. Table 10) (**B-G**).

Comment 18:

Deconvolution methods mention that all 16 are showed in figure S4. However, this doesn't seem to be the correct image. Also, there are two supplementary figures S5.

Response: We apologize for the mistake with our label, and we thank the reviewer for pointing this out. We have corrected the figure identifier in the main section of the document. The supplementary figures are now correctly labeled. The Supplemental Figure with the 16 images (Sup Fig 5, previously Sup Fig 4) is a multi-page figure.

Reviewer #2 (Remarks to the Author):

Vanrobaeys *et al.*, in this study report region specific alteration in brain transcriptome in response to sleep deprivation (SD). Their study establishes hippocampus (Hip) and neocortex (NC) as the predominant regions that show alteration in regulation of gene expression in response to SD. Surprisingly, SD drive a heterogeneous response across distinct brain regions. The unique approach of spatial transcriptomics the authors employ also reveals the alteration in the hypothalamus and thalamus in addition to Hip and NC as a consequence of SD. The body of work provides a helpful approach for the analysis of brain-section wide alterations in transcriptome and integrate it to specific anatomical regions for visualization. In addition to this the method developed by the authors will assist in identifying transcriptomic alterations, associating them to very small brain regions inaccessible with traditional methods.

The study is well conducted, discovers unique patterns of transcriptomic changes in multiple brain regions and recapitulates previous discoveries. Thus, demonstrating the sensitivity of the approach and its ability to make new discoveries and visualize them in an anatomical space. Another strength of this method is that it utilizes existing brain atlases to look at subregion-specific alterations in transcriptomics which can contribute to our understanding of region-specific responses. The development of STANLY a spatial transcriptome tool is a necessary advancement and growth in new direction for spatial transcriptomics in brain biology likely to be of great value to many researchers. The figures are excellently prepared, and the conclusions are well supported by the presented data. There are some technical clarifications and points the authors need to address to further enhance the clarity of result interpretation and better substantiate the conclusions.

Response: We thank the reviewer for this kind appraisal of our work.

Comment 1:

In figure 1D, the UMAP clusters the Allocortex and striatum like amygdalar nuclei in to two clusters. Since the same region is clustered twice, could the authors provide an explanation if this is correct or check the data to correct for this.

Response:

Yes, this is correct. The allocortex is transitional between the more evolutionarily ancient amygdala and the more evolutionarily recent neocortex and portions of the basolateral amygdala are anatomically very close to the allocortex, leading the allocortex appearing in two clusters. Transcriptionally, the BMA, the BLA and the LA share strong similarities with the other striatum-like amygdalar nuclei, including the medial amygdala subnuclei. When performing UMAP clustering, the similarity of the transcriptional signatures between the medial area (included in the striatum-like amygdalar nuclei) and the basolateral/basomedial areas leads to the basolateral/basomedial subnuclei appearing in two locations on the UMAP plot. It is important to note that we are clustering bar-coded spots and each bar-coded spot appears only once on the UMAP plot. We merged anatomically adjacent spots from the two clusters to generate the labelled brain regions.

New Text in the Revised Manuscript, p. 5:

Due to the transcriptional similarity of the basomedial and basolateral amygdalar subnuclei with the subnuclei of the amygdalar medial area, and the spatial proximity and similarity with the allocortex, UMAP clustering reports the striatum-like amygdalar nuclei and the allocortex as repeated clusters with slightly different transcriptional signatures depending upon the grouping

of the basomedial and basolateral subnuclei. We merged anatomically adjacent spots from the two clusters to generate the labelled brain regions.

Comment 2:

In the result section line 100 and 101 the authors mention that unsupervised clustering was used with transcriptional signatures, the authors should mention which of the unsupervised clustering methods was used.

Response: We used graph-based clustering for unsupervised clustering of transcriptional signatures, specifically, the Louvain clustering algorithm that includes 30 nearest neighbors and 20 principal components to identify the transcriptional signatures of each spot. This unbiased approach builds a graph where each node is a spot that is connected to its nearest neighbors in high-dimensional space. Connections are weighted based on the similarity between the spots involved, with higher weight given to spots that are more closely related. We have added additional details to the method section (page 19) to clarify the nature of the unsupervised clustering method used in this study.

New text in the Methods, p. 20-21:

A graph-based clustering was performed to identify the transcriptional signatures of each spot using the Louvain clustering algorithm that includes 30 nearest neighbors and 20 principal components. The Louvain algorithm is an unbiased approach connecting each sample spot to its nearest neighbor. The strength of the connections is weighted based on the similarity between the spots, and higher weight is given to spots more closely related. We then applied the Louvain algorithm to identify “communities” of spots that are more connected to spots in the same community than they are to spots of different communities, resulting in clustering of the sample spots.

Comment 3:

In the figure legend for fig 3 line 813, the authors seem to have used an FDR step up <0.1 while in other figures, the FDR step up is reported as <0.001 . Why was a different metric used for fig 3 specifically? This appears to be a typing error. Please elaborate.

Response: Each brain region contains a large number of spots ranging from 2000-10,000. We adopted a very conservative differential gene expression analysis using an FDR threshold of 0.001 for whole brain or large brain regions to minimize the rate of false negatives or Type II errors. However, for subregional analysis with the smaller number of spots, the risk of false negatives becomes a serious concern due to the smaller number of spots. Consequently, we lowered the FDR stringency after deconvolution of hippocampal subregions consistent to what has been used in RNA-seq studies for subregional hippocampal analysis (Chen et al 2017 PMID 28275336).

New text in the Discussion, p. 15:

Given the decreased number of sample spots in the analysis of hippocampal subregions, the FDR threshold was lowered to 0.1 for the identification of significant DEGs to reduce the number of false negatives, similar to the FDR used for RNA-seq studies of hippocampal subregions⁶¹.

Comment 4:

For the statistical analysis of digital spots (Fig. 5), a Sidak p-value correction was used. Could the authors provide a rationale for the selection of Sidak's p-value correction in the methods section. The Sidak method enhances the statistical power thus increasing the probability of false positives in comparison to more advanced/conservative methods of p-value correction.

Response:

We have taken the reviewer's suggestion to look at other p value correction methods. We chose the Sidak method because it assumes that each test is independent. We have now analyzed the data using two additional correction methods (Bonferroni and Benjamini-Hochberg methods) to assess the robustness of the results from the Sidak correction method. Bonferroni represents a conservative method correcting for the familywise error rate, i.e., the number of false positives, whereas Benjamini-Hochberg represents a stricter control for the false discovery rate, i.e., the number of false positives in ratio to the total number of positives. We have compared the results of Šidák, Bonferroni, and Benjamini-Hochberg corrections for an adjusted p-value of 0.05 and found that there are only slight differences between the results of the three approaches regarding the number of DEGs identified for whole brain analysis. The overlap of genes between the Šidák and Bonferroni correction was 100%, with the Benjamini-Hochberg identifying an additional 9 genes (S: 413 DEGs, B: 413 DEGs, BH: 422 DEGs; Supplemental Figure 12). Additionally, we've added functionality for all three correction methods to the STANLY code options.

Comment 4, part 2

In this case, the Sidak correction with the criterion that at least 3 digital spots of the 1892 might still contribute to false positives despite the new critical value after the correction. It remains unclear how the selection of 3 spots was arrived at. Therefore, we recommend that other more conservative p-value correction methods be used to control for false discoveries.

Response:

Previously, we corrected for multiple comparisons based on the number of spots, but based on this suggestion, we have now revised the p-value calculation. The new calculation corrects for transcriptome-wide significance (n=18,893), which provides a p-value as threshold for statistical significance of 2.71e-06. Because of this more stringent calculation in the revised version of our manuscript, we have since removed the 3 spot threshold. We have revised the methods to reflect the updated Sidak correction that was used. We thank the reviewer for the comments as this has allowed us to show the strength of the data set the results remain virtually the same irrespective of the method used to correct for multiple comparisons.

The following text was revised in the Methods Section, p. 28

Where α_s is the Šidák corrected p-value, α is the original p-value (and m is the number of genes in the transcriptome, n=18,893). Based on these numbers, any genes that differed between NSD and SD with a p-value < of 2.71e-06 was considered significantly differentially expressed.

The following text was added to the Methods section, p. 28:

We used the Šidák method for statistical analysis because it assumes that each test is independent of each other. However, we also tested the Bonferroni and Benjamini-Hochberg methods for FDR correction to verify the strength of the gene analysis. We found the same number of DEGs (413) for the Bonferroni correction as the Šidák, while the Benjamini-Hochberg generated 422 DEGs. The overlap of genes between the Šidák and Bonferroni correction was 100%, with the Benjamini-Hochberg including an additional 9 genes (Supplemental Figure 12). Thus, spatial transcriptomics provides a robust data set for differential gene analysis irrespective of the method used to correct for multiple comparisons.

Supplemental Figure 12

Comment 5:

Additionally, in fig 5(B1-G3) the spot comparison reduces the spatial resolution of its location, would it be possible within the scope of the project to integrate/super-impose a line map of the brain region for spatial reference.

Response: As a spatial reference, we have included an anatomical map of the brain regions in Figure 5A to assist readers in interpretation of the regions in B1 – G3. See Revised Figure 5A below. It was not feasible to superimpose line maps on B1-G3 as this obscured many of the data points.

Comment 6:

In the methods section, the authors specify that a coronal section corresponding/similar to the 45th slice in the Paxinos mouse atlas. Could the authors specify which subregions this corresponds to since the brain regions mentioned can refer to a broader anatomical region.

Response: For the purposes of tissue collection, we selected a slice that resembled slice 45 of the Paxinos atlas. The features we looked at to determine similarity were predominately the following: the shape of the dentate gyrus and fit of CA1 and CA3 bands to the atlas; size and shape of fimbria; size and separation of optic tract and internal capsule; size of lateral ventricle. We have added new text to the Methods describing the prominent tissue features that were used to determine the match to Paxinos slice 45.

New and revised text in the Methods p 19:

One coronal section per mouse, corresponding approximately to section 45 of the Paxinos Mouse Brain atlas, was mounted on Visium Spatial Gene Expression Slides (catalog no. 2000233, 10x Genomics). **For tissue collection, we selected a slice that resembled section 45 of the Paxinos Mouse Brain Atlas based on the following features: the shape of the dentate gyrus and fit of CA1 and CA3 bands to the atlas; size and shape of fimbria; size and separation of optic tract and internal capsule; size of lateral ventricle.**

Reviewer #3 (Remarks to the Author):

The study titled, "Spatial transcriptomics reveals unique gene expression changes in different brain regions after sleep deprivation", concerns a study employing newly developed spatial transcriptomic techniques and analytical methods to examine spatially localized DEGs in a coronal section of mouse brain that includes hippocampus, neocortex and subcortical regions, in response to acute sleep deprivation. The findings extend previous bulk tissue transcriptomic studies by allowing comparison of different brain regions' DEG from 13 samples (the number of samples analyzed for different aspects of the study is not always made clear). The MS is clearly written, nicely illustrated, and the study is well designed and appears to be technically sound. The expression patterns derived from the spots' transcriptomes correspond to anatomically defined brain regions. By registering ~50um diameter spots' DEGs across the samples comprising each sleep group to a common anatomical reference location, the authors identify sets of DEGs specific to localized regions. One of the most interesting findings is that most hippocampal DEGs are downregulated and the majority of neocortical DEGs are upregulated. Also, this is a technically novel application of spatial transcriptomic analysis to changes induced by sleep deprivation.

Response: We thank the reviewer for this kind appraisal of our work.

Comment:

DEGs unique to localized brain regions (hippocampal sub-regions or neocortical layers) are also identified. Although FDR appears to be well controlled, the power of the spatial transcriptomic technique for DEGs may be problematic. According to the authors, the spots may involve 10-20 different cells so that a spot may better resemble highly localized bulk tissue in that a particular gene in a given spot may be up or downregulated or change very little and this might vary considerably from spot to spot depending on the cellular makeup of the spot (for example no DEGs were identified in CA2 and CA3). Further, even with scRNAseq, using a negative binomial mixed model like NEBULA, power, estimated from realistic simulations is less than 50% for several thousands of single cells of a given cell type (many more cells than spots, each spot with mixed cells) to detect DEGs with a FC of ~1.2. Is it possible that at least some of the anatomically unique cells might be identified due to a very high rate of false negatives? Perhaps this could be more directly addressed in the discussion and the conclusions revised accordingly.

Response: We agree that the absence of DEGs in CA2 and CA3 could arise from false negatives as we did adopt a conservative approach for the analysis. As suggested by the reviewer, the lack of DEGs may arise from the lack of statistical power as there are only 210 spots total that overlay with these hippocampal subregions. Consequently, as in single-cell RNA-seq, a cluster of a hundred cells in each condition would not result in any significantly differentially expressed genes, despite our change of the FDR threshold from 0.001 for a major brain region to 0.1 for subregions such as CA2 and CA3.

As suggested by the reviewer, we have now addressed this issue more directly in the Discussion.

New text, Discussion, p. 15

Given the decreased number of sample spots in the analysis of hippocampal subregions, the FDR threshold was lowered to 0.1 for the identification of significant DEGs to reduce the number of false negatives, similar to the FDR used for RNA-seq studies of hippocampal subregions⁶¹.

New Text, Discussion, p. 16

It is probable that the small number of sample spots and subsequent lack of statistical power for analysis of the CA2 and CA3 subregions resulted in a failure to detect DEGs with some genes reported as false negatives.

REVIEWERS' COMMENTS

Reviewer #1 (Remarks to the Author):

The authors have clearly answered all the reviewers' comments and the work has benefited from the additional results, tests and clarifications.

Reviewer #2 (Remarks to the Author):

The authors have done an excellent job in preparing the revised version of their manuscript. All of my specific concerns have been addressed to my complete satisfaction. Moreover, in my opinion, the authors have also suitably addressed the comments of the other reviewers. I have no remaining concerns. The authors are to be congratulated for an excellently conducted study and well-prepared manuscript.

Graham Diering

Reviewer #3 (Remarks to the Author):

To the authors: This is a revision of the manuscript describing the study titled, "Spatial transcriptomics reveals unique gene expression changes in different brain regions after sleep deprivation." Many of the interpretations concerning sleep DEGs observed as specific to regions, cortical layers and sub-regions made by the authors, rely on the power of this technique to avoid false negatives. Yet, there is no clear assessment of the power of their spatial transcriptomic methods. This may be important since false negatives could lead to spurious conclusions concerning the anatomical specificity of differential gene expression. This kind of assessment would add greatly to the impact of this study considering that false negatives could potentially give an appearance of spatial specificity for sleep DEGs.

Specific points:

1. Within a given registered spot/group of spots defining a region/layer/subregion how many UMIs are detected/sample.
2. What is the UMI variability from sample to sample (for each kind of area that is to be analyzed rather than for the whole sample as shown in table 1)?
3. There appears to be minimal overlap of DEGs even from layer to layer in the neocortex or from hippocampus to neocortex. Would it be informative to compare UMI variability for those genes that are sleep DEGs in the regions (or layers) that are not sleep DEGs in the compared anatomical areas?

Robert Greene

A point-by-point Response to Reviewers:

We thank the reviewers and the editorial team for carefully reviewing our manuscript. We have revised the manuscript according to the comments. All the changes in the manuscript are in red.

Reviewer #1 (Remarks to the Author):

The authors have clearly answered all the reviewers' comments and the work has benefited from the additional results, tests and clarifications.

Response: We thank the reviewer for the supportive comments pointing out our noteworthy findings.

Reviewer #2 (Remarks to the Author):

The authors have done an excellent job in preparing the revised version of their manuscript. All of my specific concerns have been addressed to my complete satisfaction. Moreover, in my opinion, the authors have also suitably addressed the comments of the other reviewers. I have no remaining concerns. The authors are to be congratulated for an excellently conducted study and well-prepared manuscript.

Response: We thank the reviewer for this kind appraisal of our work.

Reviewer #3 (Remarks to the Author):

To the authors: This is a revision of the manuscript describing the study titled, "Spatial transcriptomics reveals unique gene expression changes in different brain regions after sleep deprivation." Many of the interpretations concerning sleep DEGs observed as specific to regions, cortical layers and sub-regions made by the authors, rely on the power of this technique to avoid false negatives. Yet, there is no clear assessment of the power of their spatial transcriptomic methods. This may be important since false negatives could lead to spurious conclusions concerning the anatomical specificity of differential gene expression. This kind of assessment would add greatly to the impact of this study considering that false negatives could potentially give an appearance of spatial specificity for sleep DEGs.

Specific points:

1. Within a given registered spot/group of spots defining a region/layer/subregion how many UMIs are detected/sample.

Within the neocortex, an average of 30,000 UMI counts per spot was detected across samples. This sequencing depth was maintained in a specific neocortical layer such as layer 2/3, supporting the idea that it is not differences in counts that drive the layer-specific differences in differential gene expression that we observe. We added these metrics in Supplemental Figure 12.

2. What is the UMI variability from sample to sample (for each kind of area that is to be analyzed rather than for the whole sample as shown in table 1)?

Variability from sample to sample was minimal in the neocortex, and this was consistently maintained within specific neocortical layers, such as layer 2/3. We added these metrics in Supplemental Figure 13.

3. There appears to be minimal overlap of DEGs even from layer to layer in the neocortex or from hippocampus to neocortex. Would it be informative to compare UMI variability for those genes that are sleep DEGs in the regions (or layers) that are not sleep DEGs in the compared anatomical areas?

There are 12 DEGs commonly affected after sleep deprivation in each neocortical layer. When comparing the UMI variability of these 12 DEGs with that of 12 genes unaffected by sleep deprivation in the neocortical layers, no difference is observed in UMI variability. We added these metrics in Supplemental Figure 14.

Given these findings, we would regard it as unlikely that our results are driven by false positives, but rather represent true biological changes. We have revised the “data

processing” section of the Materials and Methods to reflect the emphasis of this information.

New text in Methods, page 22:

Addressing concerns regarding the potential for false negatives, we conducted a thorough assessment of the power of our spatial transcriptomic approach. We maintained a robust sequencing depth, with an average of 30,000 UMI counts per spatial spot within the neocortex, consistently across samples and specific neocortical layers such as layer 2/3 (**Sup. Fig. 12**). Minimal variability in UMI counts was observed between samples and within the neocortex and neocortical layer 2/3, underscoring the reliability and reproducibility of our methodology (**Sup. Fig. 13**). Notably, we identified 12 DEGs common to each neocortical layer after sleep deprivation. The UMI variability of these sleep responsive DEGs did not differ from that of 12 genes unaffected by sleep deprivation in the neocortical layers (**Sup. Fig. 14**). This comparative analysis provides strong evidence against false positives, supporting the interpretation that our findings reflect genuine biological changes rather than artifacts.

We think that these changes have helped to improve the manuscript and hope that it is now suitable for publication in Nature Communications.